# Constraints on the optimization of gene product diversity

Daohan Jiang[1,2], Nevraj Kejiou[3], Yi Qiu[3], Alexander F Palazzo [ID][3✉] & Matt Pennell [ID][1,4✉]

## Abstract

**RNA and proteins can have diverse isoforms due to post-transcriptional and post-translational modifications. A fundamental question is whether these isoforms are mostly beneficial or the result of noisy molecular processes. To assess the plausibility of these explanations, we developed mathematical models depicting different regulatory architectures and investigated isoform evolution under multiple population genetic regimes. We found that factors beyond selection, such as effective population size and the number of *cis*-acting loci, significantly influence evolutionary outcomes. We found that sub-optimal phenotypes are more likely to evolve when populations are small and/or when the number of *cis*-loci is large. We also discovered that opposing selection on *cis*- and *trans*-acting loci can constrain adaptation, leading to a non-monotonic relationship between effective population size and optimization. More generally, our models provide a quantitative framework for developing statistical tests to analyze empirical data; as a demonstration of this, we analyzed A-to-I RNA editing levels in coleoids and found these to be largely consistent with non-adaptive explanations.**

**Keywords** Gene Product Diversity; Post-transcriptional Modification; Evolutionary Theory; Optimization; Constraint
**Subject Categories** Computational Biology; Evolution & Ecology; RNA Biology

## Introduction

Different RNA and protein isoforms can be expressed from the same gene, resulting in a phenomenon known as gene product diversity (Zhang and Xu, 2022). A variety of processes can generate gene product diversity, such as alternative transcription initiation (Davuluri et al, 2008; Kimura et al, 2006; Landry et al, 2003; The FANTOM Consortium and the RIKEN PMI and CLST (DGT), 2014), alternative splicing (Barbosa-Morais et al, 2012; Goldtzvik et al, 2023; Kalsotra and Cooper, 2011; Scotti and Swanson, 2016; Wright et al, 2022), alternative polyadenylation (Di Giammartino

et al, 2011), post-transcriptional RNA modifications (Farajollahi and Maas, 2010; Li and Mason, 2014; Nishikura, 2010, 2016), alternative translation initiation (Lee et al, 2012), post-translational modifications (Goldtzvik et al, 2023; Mann and Jensen, 2003), and errors during RNA or protein synthesis (de Pouplana et al, 2014; Drummond and Wilke, 2009; Dunn et al, 2013; Gout et al, 2017). The growing body of transcriptomic and proteomic data has unveiled substantial gene product diversity produced by different processes in diverse taxa, but the functional significance of the alternative isoforms remains largely unknown (Goldtzvik et al, 2023; Li and Mason, 2014; Nishikura, 2016; Wright et al, 2022; Zhang and Xu, 2022).

One explanation for observed gene product diversity is the adaptive hypothesis that the alternative isoforms perform important functions and are beneficial to the organism (de Klerk and AC't Hoen, 2015; de Pouplana et al, 2014; Liscovitch-Brauer et al, 2017). Cases of beneficial gene product modifications have been documented in various taxa. Notable examples of potentially adaptive modification events include a nonsynonymous A-to-I RNA editing event in a potassium channel protein that confers cold tolerance in polar octopuses (Garrett and Rosenthal (2012)), A-to-I editing events in filamentous fungi that fix premature stop codons in proteins involved in sexual reproduction (Liu et al, 2017; Xin et al, 2023), alternative splicing of *Sxl* transcripts that regulate sex determination in dipteran insects (Salz, 2011), and some circular RNA isoforms that function as micro RNA sponges (Hansen et al, 2013; Kristensen et al, 2019). However, such cases collectively comprise only a small portion of known gene product diversity.

An alternative view suggests that gene product diversity is largely non-adaptive and reflects errors in biochemical processes. Gene product modification processes that result in gene product diversity, like all other biochemical reactions, are fundamentally stochastic and thus prone to errors. While natural selection can act to reduce the error rate, optimization will be limited by genetic drift in a finite population. Theoretical population genetics have shown that deleterious mutations whose fitness effects are sufficiently mild given the effective population size ($N_e$) cannot be purged effectively by selection, and can accumulate in the genome over time due to mutations and genetic drift (Kondrashov, 1995; Lynch and Conery, 2003; Ohta, 1973, 1992). The effect of many molecular errors likely falls into this range, as only a limited fraction of gene product molecules are affected; as a result, selections against mutations that increase error rates can be too weak in small populations to

[1]Department of Quantitative and Computational Biology, University of Southern California, Los Angeles, CA, USA. [2]Macroevolution Unit, Okinawa Institute of Science and Technology Graduate University, Onna, Okinawa, Japan. [3]Department of Biochemistry, University of Toronto, Toronto, Canada. [4]Department of Computational Biology, Cornell University, Ithaca, NY, USA. ✉E-mail: alex.pallazo@utoronto.edu; mpennell@cornell.edu

eliminate them in the face of mutational pressure (Lynch, 2020; Lynch and Hagner, 2015). This view has been supported by analyses of various types of gene product diversity, such as alternative splicing (Bénitière et al, 2024; Pickrell et al, 2010; Saudemont et al, 2017; Xu and Zhang, 2021), alternative polyadenylation (Xu and Zhang, 2018), A-to-I RNA editing (Jiang and Zhang, 2019; Nguyen et al, 2023; Xu and Zhang, 2014), and C-to-U RNA editing (Liu and Zhang, 2018). It is also plausible that different isoforms of a gene's product are functionally equivalent, in which case the diversity per se is not adaptive even if the process that generates diversity is. That is, it is the amount of modification in a molecule rather than the precise location of any modification that matters. Processes that can potentially generate such neutral diversity include N6-methyladenosine (m6A) modification of RNA (Liu et al, 2020; Liu and Zhang, 2018; Wang et al, 2014) and protein phosphorylation (Landry et al, 2009, 2014).

Furthermore, a machinery that generates gene product diversity can be maintained by making otherwise strongly deleterious mutations reasonably benign. By restoring a proportion of gene product molecules, the gene product modification process can mitigate the negative fitness consequences of a mutation. Consequently, the modification machinery will become indispensable as its loss will reveal the deleterious effect of many past substitutions, a process known as entrenchment or "constructive neutral evolution", and has been proposed as an explanation for the increase of complexity during evolution (Lukeš et al, 2011; Muñoz-Gómez et al, 2021; Stoltzfus, 1999; Wideman et al, 2019). For example, A-to-I editing can permit G-to-A mutations as inosine (I) is recognized as guanine (G) during translation; this harm-permitting effect has likely contributed to maintenance of high A-to-I editing activity in coleoid cephalopods (clade Coleoidea, including octopuses, squids, and cuttlefishes) (Jiang and Zhang, 2019). Similarly, high C-to-U editing in plant organelles may have been entrenched after permitting T-to-C mutations (Covello and Gray, 1989; Fiebig et al, 2004; Gray, 2012).

One possible way to distinguish these alternative hypotheses in the absence of functional information for the vast majority of isoforms is to compare the observed gene product diversity within and between species to that expected under various evolutionary scenarios. However, such comparisons are not currently possible as we lack a theoretical basis for generating such expectations. While phylogenetic comparative methods have recently been applied to molecular phenotypes like gene expression levels (Chen et al, 2019; Cope et al, 2024; Dimayacyac et al, 2023; Jiang et al, 2023; Price et al, 2022), it is unclear whether conventional trait evolution models used in phylogenetic comparative analyses are suitable for modeling gene product diversity. To address these, we developed a mathematical model that connects patterns of variation in gene product diversity and the underlying evolutionary processes. In particular, we investigated two types of gene product modification processes that represent a broad range of processes that generate gene product diversity. The first type of modification simply converts an unmodified isoform to modified isoform(s) that can potentially be dysfunctional and/or toxic (Fig. 1A). Such modifications are not universally required for gene products to carry out their primary functions. Prime examples of such modifications include a variety of post-transcriptional RNA editing processes, where the RNA molecule is enzymatically modified into an alternative isoform (Farajollahi and Maas, 2010; Li and Mason,

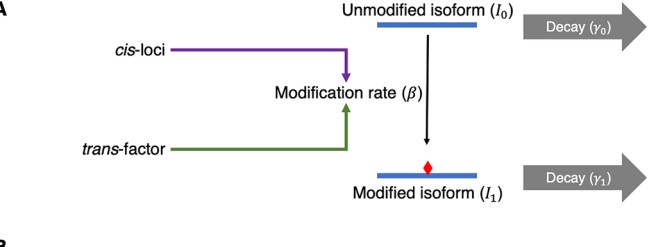

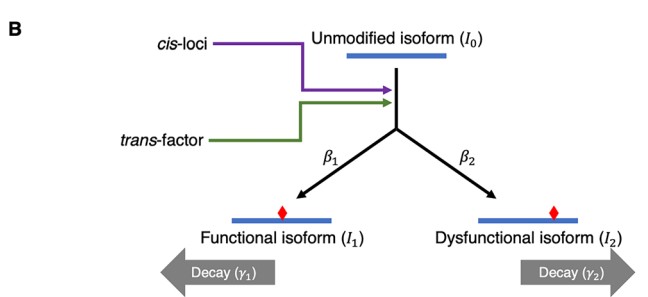

**Figure 1. Illustration of the processes leading to gene product diversity.**

A schematic illustration of editing-type (**A**) and splicing-type (**B**) gene product diversity. (**A**) An unmodified isoform ($I_0$) is enzymatically converted to a modified isoform ($I_1$). The net per-molecule conversion rate ($\beta$) is determined jointly by a *trans*-factor (enzyme performing the modification process) and a set of *cis*-loci (sequence motif underlying affinity between enzyme and substrate). (**B**) The unmodified isoform $I_0$ can be converted into either a functional isoform ($I_1$) or a dysfunctional isoform ($I_2$) through the same modification process such that two conversion rates $\beta_1$ and $\beta_2$ are affected by the same *cis*-loci and *trans*-factor.

2014; Nishikura, 2010, 2016). Thus, we will refer to this type of process as "editing-type". The second type of gene product modification process is required to produce the functional isoform, but can potentially produce mis-processed isoforms that could be dysfunctional and/or toxic (Fig. 1B). This class of modification is exemplified by RNA splicing in eukaryotes, which is generally required but can potentially produce toxic mis-spliced isoforms (Kalsotra and Cooper, 2011; Scotti and Swanson, 2016). Thus, this second type of gene product modification is referred to as "splicing-type". In both cases, each gene product modification event is regulated by a set of *cis*-loci and a *trans*-factor. Each *cis*-locus only affects a specific modification event and thus has a local effect, whereas the *trans*-factor globally affects many modification events.

Under our model, we derived phylogenetic means of the modification level under different conditions, demonstrating how the modification level is shaped by mutational pressure, genetic drift, and selection. We also investigated how opposing selection on the modification process shapes the coevolution of *cis*- and *trans*-acting loci underlying modification. At last, using computer simulations, we demonstrated that our model can recapitulate the distribution of A-to-I RNA editing levels observed in empirical studies.

# Results

## Modeling genetic architecture of isoform abundances

Under a simple model where an unmodified isoform, $I_0$, is converted to a modified isoform, $I_1$, rates at which their abundances

in the cell change over time can be written as

$$\begin{cases} \frac{dP_0}{dt} = \alpha - \beta P_0 - \gamma_0 P_0 \\ \frac{dP_1}{dt} = \beta P_0 - \gamma_1 P_1. \end{cases} \quad (1)$$

Here, $P_0$ and $P_1$ are abundances of $I_0$ and $I_1$, respectively, $\alpha$ is the rate at which $I_0$ is produced, $\beta$ is the per-molecule net rate at which $I_0$ is converted to $I_1$, and $\gamma_0$ and $\gamma_1$ are $I_0$ and $I_1$'s respective decay rates (see Table 1 for a description of all model parameters). An equilibrium is reached when both rates are equal to zero:

$$\begin{cases} \alpha - \beta P_0 - \gamma_0 P_0 = 0 \\ \beta P_0 - \gamma_1 P_1 = 0. \end{cases}$$

Solving the system of equations gives equilibrium isoform abundances:

$$\begin{cases} P_0 = \frac{\alpha}{\beta + \gamma_0} \\ P_1 = \frac{\alpha\beta}{\gamma_1(\beta + \gamma_0)}. \end{cases} \quad (2)$$

The same modeling approach can be generally applied to systems with more isoforms (see "Methods").

In our model, the per-molecule conversion rate $\beta$ is controlled by a *trans*-factor (an enzyme that performs gene product modification) and a set of *cis*-loci (genomic loci encoding regions adjacent to the site subject to modification that affect binding affinity between the gene product molecule and the *trans*-factor). The *trans*-factor's effect on $\beta$ is characterized by a *trans*-genotypic value, $Q$, which reflects the modification enzyme's expression level and/or catalysis efficiency. The *cis*-genotype's effect is summarized by a normalized *cis*-genotypic value $\hat{v}$. A high $\hat{v}$ indicates strong binding between the modification enzyme and the substrate, which results in high modification efficiency, whereas a low $\hat{v}$ means weak enzyme-substrate binding and low modification efficiency. Each *cis*-locus can have either an *effector* allele that facilitates enzyme binding, or a *null* allele that has no effect. In this study, we focused on a simple model where all loci's effector alleles have an equal, additive effect (Lynch, 2020), so $\hat{v}$ is calculated as $\hat{v} = v/l$, where $l$ is the number of *cis*-loci that affect the modification and $v$ is the total number of effector alleles. This model can readily be extended to incorporate variation in the contribution of different loci—for example, a skewed distribution where one locus has major effect while others' effects are much weaker.

Given values of $Q$ and $\hat{v}$, $\beta$ is calculated as

$$\beta = Q(C\hat{v} + \epsilon) \text{ where } C > 0, \epsilon \geq 0. \quad (3)$$

Here, $C$ represents whole-molecule features that modulate the *cis*-loci's effect size, such as the secondary structure of RNA or protein, and $\epsilon$ is the rate of nonspecific modification (promiscuous activity of the enzyme independent of the *cis*-genotype).

For editing-type modification, we focused on a simple scenario where two isoforms, the unmodified isoform $I_0$ and modified isoform $I_1$, are present (Fig. 1A); the generic, two-isoform model described above is thus readily applicable. We considered values of $l$ that are relatively small ($\leq 10$), as empirical studies suggest that sequence motifs with major effects on RNA modifications usually

**Table 1. Definitions and notations of parameters.**

| Parameter | Definition |
|---|---|
| $I_i$ | The $i$th modified isoform; $I_0$ represents the unmodified isoform. |
| $P_i$ | Abundance of $I_i$. |
| $\alpha$ | Rate at which $P_0$ is produced. |
| $\beta_i$ | Per-molecule net rate at which the $I_0$ is converted to the $I_i$. |
| $\gamma_i$ | Decay rate of $I_i$. |
| $f$ | Modification level; $f = \frac{P_1}{P_1 + P_0}$ for editing-type and $f = \frac{P_2}{P_1 + P_2}$ for splicing-type. |
| $l$ | Number of *cis*-loci affecting $\beta$. |
| $v$ | *cis*-genotypic value characterizing the combined effect of the *cis*-genotype on $\beta$. |
| $v_{max}$ | Value of $v$ when every locus has an effector allele. |
| $\hat{v}$ | Normalized *cis*-genotypic value, $\hat{v} = \frac{v}{v_{max}}$. |
| $Q$ | *trans*-genotypic value underlying $\beta$. |
| $C$ | Parameter characterizing gene-level feature that affect *cis*-loci's effect size on $\beta$. |
| $\mu_{01}$ | Mutation rate from null allele to effector allele per *cis*-loci. |
| $\mu_{10}$ | Mutation rate from null allele to effector allele per *cis*-loci. |
| $\omega$ | Overall fitness. |
| $\omega_i$ | Fitness with respect to $P_i$. |
| $\sigma_i$ | Width of the fitness function when $P_i$ is under stabilizing selection. |
| $\lambda_i$ | Parameter characterizing speed at which $\omega_i$ declines with $P_i$ when $I_i$ is toxic. |
| $s$ | Coefficient of selection of a mutation. |
| $N_e$ | Effective population size. |
| $\Pr(i \rightarrow j)$ | Probability that $v$ changes from $i$ to $j$ via a substitution in a time step. |
| $\mathbf{T}$ | Transition matrix for the genotypic value $v$. |
| $\mathbf{v}_t$ | Probability distribution of $v$ at a given time $t$. |
| $v_t$ | Value of $v$ at a given time $t$; $v_0$ is the starting value. |
| $U_Q$ | Rate of mutations affecting $Q$. |
| $S_Q$ | Standard deviation of mutation's effect on $\ln Q$. |
| $\sigma_Q$ | Width of the fitness function of $Q$. |

consist of a small number of nucleotide sites (Farajollahi and Maas, 2010; Lehmann and Bass, 2000; Li and Mason, 2014). In an extreme case, A-to-I editing in filamentous fungi, the nucleotide site immediately upstream the editable A site appears to be the only *cis*-locus, where the effector allele is a T base (Liu et al, 2016, 2017; Wang et al, 2016).

For splicing-type modification, we considered a model where the unmodified isoform $I_0$ is converted to two modified isoforms, a functional isoform $I_1$ and a dysfunctional isoform $I_2$, at rates $\beta_1$ and $\beta_2$, respectively. As $I_1$ and $I_2$ are essentially products of the same process, their respective modification rates $\beta_1$ and $\beta_2$ are controlled by the same *cis*-loci (Fig. 1B); thus, we assumed an allele that does not facilitate the production of the $I_1$ will facilitate the production of $I_2$ and vice versa. For convenience, the *cis*-genotypic value is defined as the *cis*-genotype's effect on $\beta_1$ for splicing-type modification. Hence:

$$\beta_1 = Q(C\hat{v} + \epsilon)$$

and

$$\beta_2 = Q(C(1 - \hat{v}) + \epsilon).$$

As the splicing of a gene's transcript can be affected by a relatively large number of loci, including splicing enhancers, inhibitors, and cryptic splice sites (Wang et al, 2005; Wang and Burge, 2008), we considered relatively large values of $l$ (10, 20, 30, 40, and 50) for splicing-type modification. We assumed $\gamma_0 = 0$ but a high $Q$ such that $I_0$ only comprise a small fraction of the gene product (that is, $P_0/(P_0 + P_1 + P_2) \approx 1\%$) to recapitulate the fact that splicing occurs co-transcriptionally (Herzel et al, 2017). We also had $\gamma_2$ significantly greater than $\gamma_1$ to reflect the effect of quality-control processes, such as nonsense-mediated RNA decay (Frischmeyer and Dietz, 1999; Kurosaki and Maquat, 2016; Kurosaki et al, 2019), or nuclear retention and decay of intronic polyadenylated transcripts mediated by recognition of intact 5′ splice site (Lee et al, 2015, 2022, 2025). The model for splicing-type modification can be readily applied as long as gene product diversity results from alternative products of an indispensable process in gene expression. For instance, it may be applied to alternative polyadenylation, in which case $I_0$ represents nascent RNA, and $I_1$ and $I_2$ represent RNAs polyadenylated at different sites.

## Evolutionary scaling of mean modification level

In the cases where the only loci that evolve are the *cis*-loci, which could occur if the *trans*-factor is invariable because of its pleiotropic effects, and the *cis*-loci's fitness effect is only mediated by gene product modification, the evolution of the *cis*-genotypic value $v$ can be modeled as a discrete-state Markov process. Consequently, we can derive the probability distribution of $v$ (and $\hat{v}$) given the initial distribution and regime of selection after evolution for a given amount of time (Lynch, 2020; Lynch and Hagner, 2015). We then asked what the expected relative abundance of a dysfunctional, toxic isoform—for example, one that reduces fitness due to mis-interactions with other biomolecules—will be in the face of mutation, drift, and selection.

For editing-type modification, we considered a deleterious modification event that converts an unmodified isoform $I_0$ that is functional to a modified isoform $I_1$ that is not functional but toxic. That is, $P_0$ is under stabilizing selection and fitness with respect to $P_0$ is a Gaussian function of $\ln P_0$:

$$\omega_0 = \exp\left(-\frac{\ln P_0 - \ln \tilde{P}_0}{2\sigma^2}\right),$$

where $\tilde{P}_0$ is the optimal value of $P_0$ and $\sigma$ is width of the fitness function. Fitness with respect to $P_1$, in contrast, declines with $P_1$:

$$\omega_1 = \exp(-\lambda P_1),$$

where $\lambda$ is a parameter characterizing the level of toxicity. Together, the overall fitness is given by

$$\omega = \omega_0\omega_1 = \exp\left(-\frac{\ln P_0 - \ln \tilde{P}_0}{2\sigma^2}\right) \cdot \exp(-\lambda P_1). \qquad (4)$$

The phenotype of interest we examined was the modification level, $f = P_1/(P_0 + P_1)$. For each combination of parameter values, we calculated the mean of $v$ after evolution from $v = 0$ for $10^8$ time steps and the corresponding $f$, which we refer to as a phylogenetic mean of modification level (mean modification level, for short). Under all conditions examined, the mean modification level declines with effective population size $N_e$ (Fig. 2). Mutational bias towards the effector allele makes the mean modification level higher, whereas bias in an opposite direction makes it lower (Fig. 2A–C). For a given $N_e$ and the per-locus mutation rate, the mean modification level becomes higher when the number of *cis*-loci, $l$, is high, which is most pronounced at relatively small $N_e$ (Fig. 2A–C). This relationship between modification $l$ is explained by the relative size of genotypic space that produce the optimal phenotype. The optimal genotype, which leads to $v = 0$, corresponds to $2^{-l}$ of the genotypic space. Thus, when $l$ is large, it is harder to maintain an optimal genotype in the face of mutational pressure towards non-zero *cis*-genotypic values when $l$ is greater (Lynch, 2020). Another key factor affecting the mean modification level is expression level of the gene (i.e., optimal $P_0$, reached when $\beta = 0$): mean modification level is lower when the gene is more highly expressed (Fig. 2D–F). This relationship is driven primarily by the toxic isoform's abundance, $P_1$ instead of $P_0$—given the modification level, there will be higher $P_1$ and thus greater fitness cost mediated by toxicity when the gene is highly expressed.

For splicing-type modification, it was a scenario where $I_1$ is functional and $I_2$ is toxic that was considered. Thus, the overall fitness is given by

$$\omega = \omega_1\omega_2 = \exp\left(-\frac{\ln P_1 - \ln \tilde{P}_1}{2\sigma^2}\right) \cdot \exp(-\lambda P_2), \qquad (5)$$

where notations follow those in Eq. (4). Modification level in this case is defined as the relative abundance of the dysfunctional and toxic isoform $I_2$ out of all modified products, $f = P_2/(P_1 + P_2)$. As in the case of editing-type modification, mean level of splicing-type modification also declined with $N_e$ and gene expression level, and increased with $l$ (Fig. EV1). We also examined the effect of a quality-control mechanism like nonsense-mediated decay (i.e., high $\gamma_2$) and confirmed that faster decay of $I_2$ can substantially lower the modification (Fig. EV1). When it is the *cis*-genotypic value instead of the modification level that is under concern, results under different values of $\gamma_2$ are mostly similar (Fig. EV2). Together, we show that a quality-control mechanism (i.e., high $\gamma_2$) can have a harm-permitting effect by making *cis*-mutations less deleterious, thereby increasing their fixation probability: with high $\gamma_2$, the harmful effect of producing high level of $I_2$ is reduced, so genotypes that encode high $\beta_2$ can be permitted.

## Non-monotonic scaling in *cis-trans* coevolution

Given that non-adaptive gene product diversity will be present when selection is unable to optimize the *cis*-loci in the face of mutational pressure and genetic drift, obvious questions are: why did this machinery evolve in the first place, and how is this maintained? These are particular pertinent for editing-type modifications that are not an indispensable part of gene expression. Presumably, such gene product modification processes must have

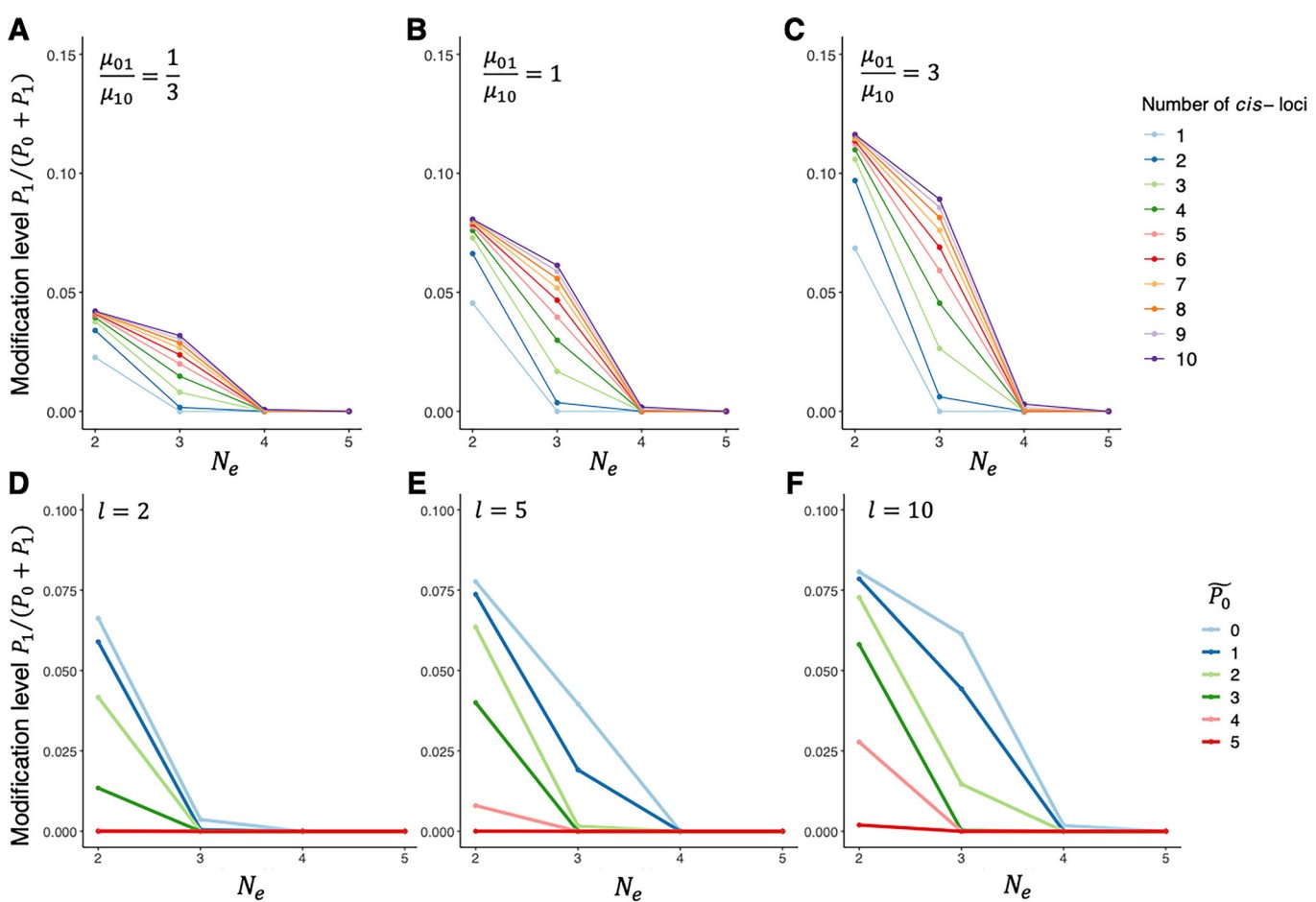

**Figure 2. Mean modification level varies with population genetic environment and genetic architecture.**

Scaling between mean modification level of a deleterious editing-type modification to effective population size $N_e$ (shown in log10 scale). (A–C) Response of mean modification level to $N_e$ given different combinations of *cis*-loci number ($l$) and mutation rates ($\mu_{01}$, $\mu_{10}$), with optimal expression level $\bar{P}_0 = \exp(1)$ ($\ln \bar{P}_0 = 1$). (A) Mutational bias is towards the null allele that does not facilitate modification. (B) Mutations of two directions have equal mutation rates. (C) Mutational bias is towards the effector allele that facilitates modification. (D–F) Response of mean modification level to $N_e$ given different $\bar{P}_0$ with $l = 2$ (D), $l = 5$ (E), and $l = 10$ (F) in the absence of mutational bias. All results are derived with initial *cis*-genotypic value $v_0 = 0$, time of evolution $T = 10^8$ time steps, total mutation rate per *cis*-locus $\mu = \mu_{01} + \mu_{10} = 2 \times 10^{-9}$, $Q = 1$, $\gamma_0 = 1$, and $\gamma_1 = 1$. The optimal expression level $\bar{P}_0$ is set to be equal to $P_0$ in the absence of modification (i.e., $\bar{P}_0 = a/\gamma_0$) in all cases.

additional essential functions unrelated to the set of modification events studied here, such that loss or suppression of the modification machinery will have a strongly deleterious effect. This additional function can be interpreted either as unrelated to the type of modification under concern, or as an additional set of modification event(s) of the type under concern that are beneficial. For instance, if the type of modification under concern is nonsynonymous RNA editing, this additional function could be interpreted as editing of non-coding RNAs, or as a set of beneficial nonsynonymous editing events.

To better understand evolutionary dynamics when the modification machinery is under opposing selection forces, we considered a scenario where modification events under concern are deleterious, but the *trans*-genotypic value $Q$ is under stabilizing selection due to its contribution to an additional fitness component (Fig. 3A; see also "Methods"), and conducted simulations to investigate how *cis*- and *trans*-acting loci will respond to selection. We simulated evolution under different combinations of $N_e$, $l$, and

strength of selection on $Q$. The simulation started from a high value of $Q$ and intermediate *cis*-genotypic values (i.e., values with the largest corresponding genotypic space), representing a state that high modification activity had just evolved and optimization of *cis*-loci have not yet started.

We found the among-lineage average of $Q$ at the end of the simulation, denoted $\overline{Q}$, is generally higher when selection on $Q$ is strong (Fig. 3B–D, red versus blue curves). Critically, the relationship between $\overline{Q}$ and $N_e$ is not monotonic: $\overline{Q}$ first decreases with $N_e$, but increases when $N_e$ is sufficiently large. Such a relationship indicates different modes of optimization at different $N_e$. When $N_e$ is too small, neither *cis*- nor *trans*-genotypic values can be efficiently optimized, so the starting condition is mostly maintained; when $N_e$ is intermediate, as selection is still not efficient enough to optimize *cis*-loci of individual modification events in the face of mutational pressure and genetic drift, relatively low $Q$ evolves to reduce the deleterious effect of gene product modifications globally. When $N_e$ is sufficiently large, selection can have the

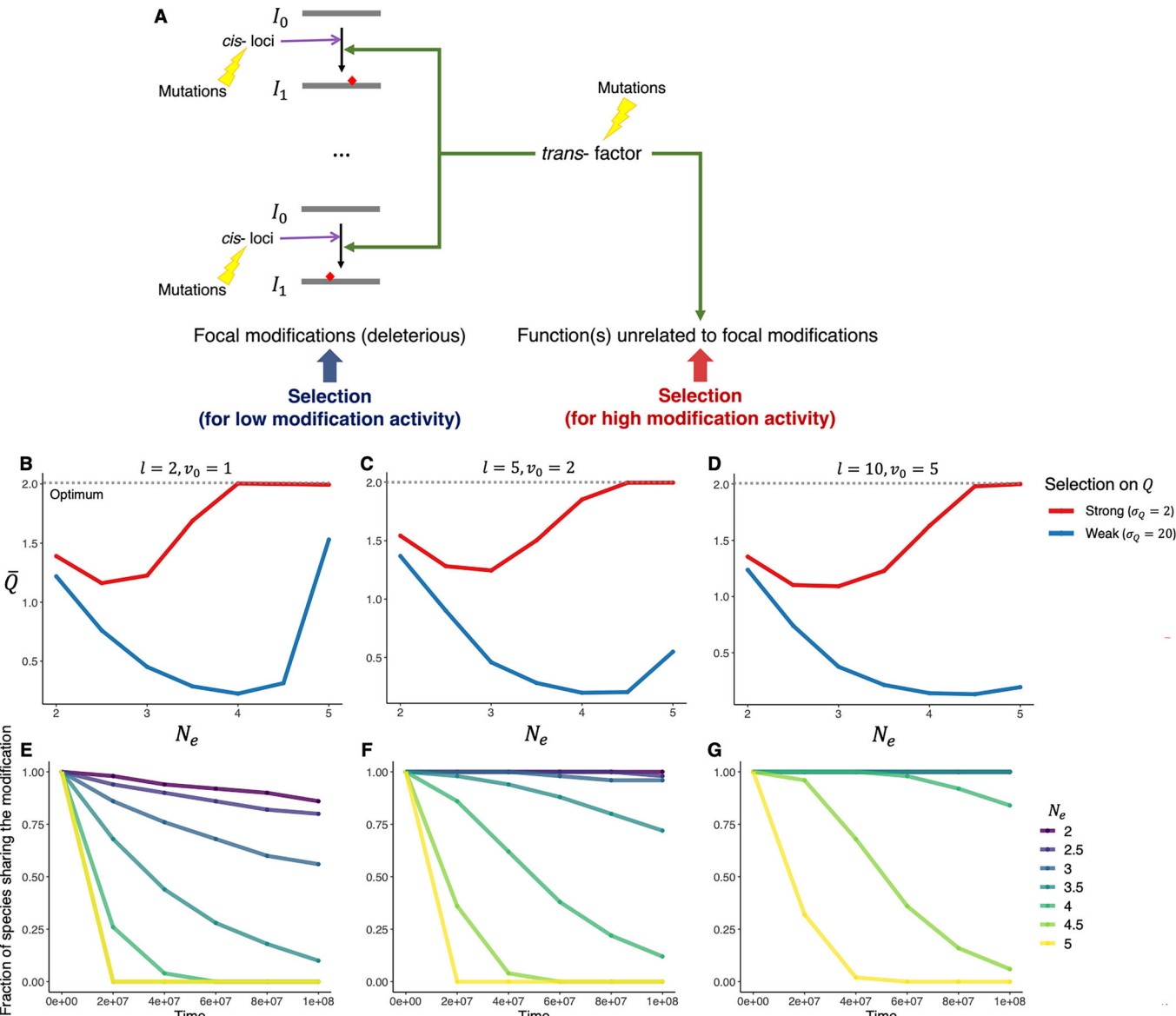

**Figure 3. Coevolution of *cis-* and *trans*-acting loci when the gene product modification machinery is under opposing selection forces.**

(A) Schematic illustration of the scenario. The *trans*-factor, while causing a number of deleterious editing-type modification events (focal modifications), also performs an essential function independent of the focal modifications. Selection against deleterious modification may act to reduce the *trans*-genotypic value ($Q$), while selection mediated by the other function(s) act to maintain an optimal value of $Q$ ($\tilde{Q}$). (B–D) Non-monotonic response of mean of $Q$ across lineages to $N_e$ (shown in log10 scale) with $Q$ under stabilizing selection and 100 genes subject to deleterious modification. Curves of different colors correspond to scenarios of strong (red) and weak (blue) selection on $Q$. Optimum of $Q$ is denoted by the dashed line. All simulations started with an intermediate *cis*-genotypic value with the largest corresponding genotypic space. (E–G) Sharing of modification events over time. $Y$ axes represent the among-gene median of proportion of lineages (species) that share a modification event when selection on $Q$ is strong ($\sigma_Q = 2$). When two curves in the same panel completely overlap, the one with the largest corresponding $N_e$ is shown. In (B, E), $l = 2$ and $v_0 = 1$; in (C, F), $l = 5$ and $v_0 = 2$; in (D, G), $l = 10$ and $v_0 = 5$.

population approach the global optimum where $Q$ is optimal and modification at individual sites are optimized locally via *cis*-substitutions.

The above interpretation predicts that the tipping point where $\overline{Q}$ starts to increase with $N_e$ should correspond to a smaller $N_e$ when selection on $Q$ is stronger, and that $\overline{Q}$ will be lower, for a given $N_e$, when mutational pressure is strong (i.e., when $l$ is large) and *cis*-loci are harder to optimize. Both predictions are confirmed by our simulations (Fig. 3B–D). The tipping point occurs at about

$N_e = 10^{2.5}$ or $N_e = 10^3$ when selection on $Q$ is strong (width of fitness function $\sigma_Q = 2$; see "Methods"), but at about $N_e = 10^4$ when selection on $Q$ is weak ($\sigma_Q = 20$). In addition, when $l$ is large, $\overline{Q}$ increases less with $N_e$ after the tipping point (Fig. 3B–D).

We also examined how the deleterious modification events are shared across lineages over time. For each modification event, we calculated the fraction of lineages that shared it, and used the median across all 100 modification events to represent the level of conservation given the parameter combination (see "Methods").

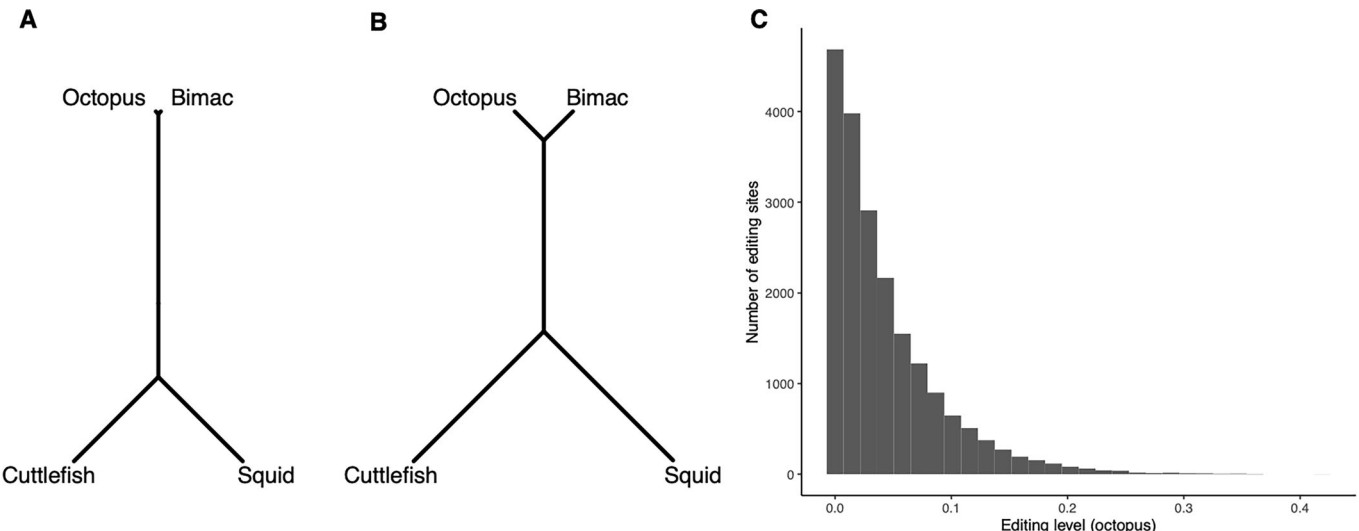

**Figure 4. Simulations of A-to-I RNA editing along the coleoid phylogeny.**

Evolutionary simulations recapitulated patterns of A-to-I RNA editing in four coleoid species, the octopus (*Octopus vulgaris*), the bimac (*O. bimaculoides*), the squid (*Doryteuthis pealeii*), and the cuttlefish (*Sepia oficianalis*). (**A**) Phylogenetic tree of four coleoid species. (**B**) Neighbor-joining tree of four coleoid species based on simulated editing levels. An unrooted version is shown in (**A**) as it is readily comparable to (**B**). (**C**) Distribution of editing levels across genes in the octopus.

The fraction of lineages sharing the modification generally declined over time but declined more rapidly when $N_e$ is large and when $l$ is small (Figs. 3E–G and EV3). When $N_e$ is relatively small (e.g., $N_e < 10^3$) and/or $l$ is high (e.g., $l \geq 10$), modifications are shared by a large proportion of, and in some case, all lineages (Figs. 3E–G and EV3).

## Simulated data recapitulate divergence of A-to-I RNA editing in coleoids

To complement our theoretical results, we asked whether simulation under our model is able to generate a distribution of modification levels that is similar to those observed in empirical studies. To this end, we examined if simulations could recapitulate the distribution of A-to-I RNA editing levels in coleoids. For this group, we cannot yet test the relationship between editing and $N_e$ because data of editing is are only available for four species and we would lack statistical power. However, editing levels of a large number of editing sites have been reported in each species, allowing us to examine the distribution of editing levels across sites. Previous studies reported preponderant A-to-I editing by the ADAR family of enzymes (adenosine deaminases acting on RNA) in four coleoid species' neural tissues (Alon et al, 2015; Liscovitch-Brauer et al, 2017), which results from less restricted cellular localization of ADAR (Vallecillo-Viejo et al, 2020). The distribution of editing levels at coding sites is strongly skewed, with a vast majority of editing sites having rather low (<1%) editing levels (Alon et al, 2015; Jiang and Zhang, 2019; Liscovitch-Brauer et al, 2017). We simulated evolution of 20,000 editing-type modification events, including 10,000 neutral modifications and 10,000 deleterious modifications along a phylogenetic tree of four coleoid species (Fig. 4A), with some gene-specific parameters ($\alpha$, $l$, and $C$) sampled from pre-specified distributions. To reproduce a skewed distribution of modification levels like those observed in empirical studies (Alon et al, 2015; Jiang and Zhang, 2019; Liscovitch-Brauer et al, 2017), we sampled $C$ from an exponential distribution with a moderate mean (i.e., magnitudes higher than $\epsilon$ but not high enough to produce an editing level above 10%). Editing levels from our simulation showed a strong phylogenetic signal; the neighbor-joining tree based on distance in editing levels recapitulates the topology of the species tree and relative branch lengths; Fig. 4B). Furthermore, there is a skewed distribution of editing levels in each species (exemplified by distribution in octopus shown in Fig. 4C). Similar patterns were seen when neutral (Figs. 5A,B) and deleterious (Fig. EV4C,D) editing sites were examined separately, though deleterious editing levels are generally lower and the distribution of editing levels is more skewed.

## Discussion

In this study, we developed a theoretical model for the evolution of gene product diversity, investigating how the interplay of mutations, genetic drift, and selection on isoform abundances shapes evolutionary dynamics. Our analyses reveal that the optimization of gene product diversity can be highly constrained by the underlying genetic architecture, effective population size, gene expression levels, and pleiotropic effects of the gene product modification machinery. These constraints suggest that a substantial portion of observed gene product diversity is likely to be evolutionarily sub-optimal rather than adaptive.

We find that when selection is too weak relative to mutational pressure and genetic drift, populations will maintain deleterious modifications even over evolutionary timescales. The model consistently shows that mean modification levels decline with increasing effective population size, which we would expect in scenarios where they tend to be deleterious. This pattern is apparent across both editing-type and splicing-type modifications, indicating a general principle in the evolution of gene product diversity. These findings align with previous empirical observations across various types of gene product diversity, such as the negative

correlation between $N_e$ and the overall rate of alternative splicing observed across metazoan species (Bénitière et al, 2024).

The effect of the number of *cis*-acting loci ($l$) on modification levels is particularly noteworthy, with higher $l$ values resulting in higher deleterious modification levels, especially under small population sizes. This relationship can be explained by considering the relative size of genotypic space that produces the optimal phenotype. When $l$ is large, maintaining an optimal genotype becomes increasingly difficult in the face of mutational pressure, as the optimal genotype corresponds to only $2^{-l}$ of the total genotypic space (Lynch, 2020). This theoretical prediction provides a testable hypothesis for future comparative analyses of gene product diversity.

Another key factor affecting modification levels is gene expression level, with more highly expressed genes displaying lower modification levels. This pattern is driven primarily by the toxic isoform's abundance-given the same modification level, a highly expressed gene will produce more toxic isoforms, resulting in greater fitness costs. This relationship between expression level and purifying selection has been observed across multiple molecular phenotypes, including various types of gene product modifications (Bénitière et al, 2024; Pickrell et al, 2010; Saudemont et al, 2017; Xu and Zhang, 2018) but also those of sequence evolution (Managadze et al, 2011; Zhang and Yang, 2015), an expression level (Liao and Zhang, 2006), and translation fidelity (Mordret et al, 2019).

Our simulations of *cis-trans* coevolution reveal particularly interesting dynamics when the gene product modification machinery experiences opposing selection pressures. The resulting non-monotonic relationship between $N_e$ and the overall editing activity (characterized by *trans*-genotypic value $Q$) indicates different outcomes at different population sizes. When $N_e$ is very small, neither *cis*- nor *trans*-acting loci can be efficiently optimized, leading to maintenance of the starting condition. At intermediate $N_e$, $Q$ evolves to be relatively low to globally reduce deleterious modifications, while at larger $N_e$, selection approaches the global optimum where $Q$ reaches its optimal value and modification at individual sites is optimized locally via cis-substitutions. This theoretical prediction is consistent with previous findings on global versus local optimization in the evolution of quality-control mechanisms (Ho and Hurst, 2021; Koonin, 2006, 2016; Rajon and Masel, 2011; Xiong et al, 2017). In actual biological systems, the global solution may manifest as lowered expression or catalytic efficiency of the *trans*-factor, or an auto-regulatory mechanism where the *trans*-factor modifies its own gene product and trigger negative regulatory effects when its expression is too high (Carvill and Mefford, 2020; Lareau et al, 2007; Lee et al, 2025; Ni et al, 2007).

Importantly, our results are consistent with the idea that gene product diversity is maintained due to pleiotropic functions of the molecular machinery that generates it. For instance, A-to-I RNA editing has been implicated in preventing autoimmune responses by modifying transcripts from repetitive elements (Chung et al, 2018; de Reuver et al, 2022; Karki et al, 2021; Liddicoat et al, 2015) and suppressing retrotransposition (Orecchini et al, 2017). The unusually high A-to-I editing activity observed in coleoid neural tissues may serve similar functions, as editing is enriched in repetitive elements in these species (Albertin et al, 2022). Similarly, m6A modification appears involved in repression of endogenous retroviruses (Chelmicki et al, 2021) and decay of mis-processed RNA (Lee et al, 2025) through mass-action mechanisms. Such functions may explain the evolutionary persistence of modification processes that otherwise appear to generate predominantly non-adaptive diversity.

Our simulations also demonstrated that when $N_e$ is relatively small or when $l$ is large, modifications are often shared across divergent lineages even when they are deleterious. This finding has important implications for interpreting phylogenetic conservation of gene product modifications. While conservation has traditionally been interpreted as evidence for adaptation (Xu and Zhang, 2015), our results suggest that phylogenetic conservation alone is insufficient to infer adaptive value. For individual modification events, functional evidence beyond mere conservation is necessary to support an adaptive hypothesis. This is further backed up by our simulation of the evolution of A-to-I RNA editing in coleoids. We successfully recapitulated the empirical distribution of editing levels, with the majority showing low editing frequencies and a strong phylogenetic signal. This suggests that observed patterns of gene product diversity can be explained by a relatively simple non-adaptive model where whole-molecule binding affinity follows a skewed distribution. The difference between neutral and deleterious editing events in our simulations is consistent with previous observations that the distribution of editing levels at diversifying sites is more skewed than that of synonymous sites (Jiang and Zhang, 2019). Because data of editing is only available for four coleoid species, we cannot yet test if patterns of editing is correlated with $N_e$ in this group like what was done for alternative splicing (Bénitière et al, 2024). Such a test could be done in future studies as data of editing in more coleoid species as well as matching $N_e$ estimates become available. It is worth noting at last that the four coleoid species for which editing sites have been identified do differ in overall abundance of coding RNA editing (Jiang and Zhang, 2019; Liscovitch-Brauer et al, 2017), and it would be interesting to test if this is indeed explainable by variation in $N_e$.

While our model does not exclude the possibility of adaptive modifications evolving secondarily with the modification machinery already in place, it is compatible with a model of constructive neutral evolution (Lukeš et al, 2011; Muñoz-Gómez et al, 2021; Stoltzfus, 1999; Wideman et al, 2019) where deleterious substitutions can be permitted and entrenched while the modification machinery is maintained due to its additional function. Modifications that restore the permitted substitutions can also be considered as a latent function that contributes to the modification process's maintenance.

In addition to the fraction of gene product modification events that are adaptive, the overall distribution of their fitness effects is also of great interest, but yet generally unknown. In the case of A-to-I editing, as its effect on RNA or protein sequences is equivalent to that of A-to-G mutations, it is intuitive to expect that the distribution of fitness effects (DFE) of A-to-I editing events is similar to that of A-to-G mutations, though the magnitude of fitness effect of editing is likely smaller as each editing event affects only a fraction of transcripts whereas each mutation affects all RNA molecules transcribed from the mutated copy of gene. Although most individual editing events' fitness effects are unknown, the similarity between effects of editing events and mutations that cause the same amino acid change has indeed been shown in empirical studies of editing events with major effects (Birk et al, 2023; Higuchi et al, 2000). Studies of DFE of non-lethal spontaneous mutations, which are mostly point mutations, have revealed that there are many more deleterious mutations than beneficial ones, and that most of the deleterious mutations have weak effects (Eyre-Walker and Keightley, 2007). Hence, a model where A-to-I editing events are mostly neutral or deleterious is

likely to be consistent with the real DFE. Other gene product modifications whose effects on RNA or protein sequences are equivalent to those of point mutations—for example, C-to-U editing, whose effect resembles that of C-to-T mutations—are also likely to have similar DFEs. The effect of other types of modifications on gene products, on the other hand, are not necessarily comparable to mutations; for example, mis-splicing can result in the inclusion of intronic sequences in the transcript (Barbosa-Morais et al, 2012; Goldtzvik et al, 2023; Kalsotra and Cooper, 2011; Scotti and Swanson, 2016; Wright et al, 2022) or production of circular RNAs (Kristensen et al, 2019). Nevertheless, as such errors cause even greater disturbance to the gene product's molecular structure, it is likely they are generally more deleterious than alterations of individual nucleotide or amino acid sites.

It is worth noting that models of editing-type and splicing-type modifications examined in this study, while flexible enough for modeling a broad range of processes that generate gene product diversity, may not be well suited for others. For instance, the use of alternative promoters or transcription initiation sites can also produce gene product diversity (Davuluri et al, 2008; Kimura et al, 2006; Landry et al, 2003; The FANTOM Consortium and the RIKEN PMI and CLST (DGT, 2014). Such diversity cannot be properly modeled as editing-type or splicing-type and would require different versions of the model (see also "Methods"). The evolutionary dynamics of these additional mechanisms represent an important area for future investigations.

Looking forward, a critical impediment to more comprehensive empirical analyses is the lack of appropriate statistical phylogenetic tests for comparing observed distributions of gene product diversity with theoretical expectations. While standard statistical approaches for quantitative traits have proven adequate for modeling mRNA abundance evolution (Chen et al, 2019; Dimayacyac et al, 2023), enabling direct theory-data comparisons (Cope et al, 2024; Price et al, 2022), these approaches may not be suitable for gene product diversity due to its unique genetic and mutational architecture. Our model provides a quantitative framework for developing such statistical tests.

# Methods

Reagents and Tools Table

| Reagent/resource | Reference or source | Identifier or catalog number |
|---|---|---|
| **Experimental models** | | |
| **Recombinant DNA** | | |
| **Antibodies** | | |
| **Oligonucleotides and other sequence-based reagents** | | |
| **Chemicals, enzymes and other reagents** | | |
| **Software** | | |
| R | https://www.r-project.org | |
| **Other** | | |

## Isoform abundances at equilibrium

Let us consider a scenario where an unmodified isoform (denoted $I_0$) is converted to a modified isoform (denoted $I_1$). Their abundances are denoted $P_0$ and $P_1$, respectively.

The rate at which $P_0$ changes through time is given by

$$\frac{dP_0}{dt} = \alpha - \beta P_0 - \gamma_0 P_0, \tag{6}$$

where $\alpha$ is the rate at which the unmodified isoform is produced, $\beta$ is the net conversion rate from $I_0$ to $I_1$, and $\gamma_0$ is the unmodified isoform's decay rate.

The rate at which $P_1$ changes through time is given by

$$\frac{dP_1}{dt} = \beta P_0 - \gamma_1 P_1, \tag{7}$$

where $\gamma_1$ is the modified isoform's decay rate.

An equilibrium is reached when

$$\begin{cases} \frac{dP_0}{dt} = \alpha - \beta P_0 - \gamma_0 P_0 = 0 \\ \frac{dP_1}{dt} = \beta P_0 - \gamma_1 P_1 = 0. \end{cases} \tag{8}$$

Solving the above system of equations gives

$$\begin{cases} P_0 = \frac{\alpha}{\beta+\gamma_0} \\ P_1 = \frac{\alpha\beta}{\gamma_1(\beta+\gamma_0)}. \end{cases} \tag{9}$$

The proportion of the gene product that is modified is

$$f = \frac{P_1}{P_0 + P_1} = \frac{\beta}{\beta + \gamma_1}. \tag{10}$$

The same model can be extended to more complex cases where more isoforms of the same gene's product are present. If $n$ unique isoforms ($I_1$, …, $I_n$) can be produced by modifying $I_0$ and each molecule of $I_0$ can only be modified into one alternative isoform, the equilibrium is reached when

$$\begin{cases} \frac{dP_0}{dt} = \alpha - (\sum_{i=1}^{n}\beta_i)P_0 - \gamma_0 P_0 = 0 \\ \frac{dP_1}{dt} = \beta_1 P_0 - \gamma_1 P_1 = 0 \\ \dots \\ \frac{dP_n}{dt} = \beta_n P_0 - \gamma_n P_n = 0. \end{cases} \tag{11}$$

In this case, $\beta_1$, …, $\beta_n$ are net rates at which $I_0$ is converted to $I_1$, …, $I_n$, respectively, and $\gamma_1$, …, $\gamma_n$ are decay rates of $I_1$, …, $I_n$. The above system of equations can be rearranged and written in a matrix ($\mathbf{A}x = \mathbf{b}$) form:

$$\begin{bmatrix} \sum_{i=1}^{n}\beta_i + \gamma_0 & 0 & \dots & 0 \\ \beta_1 & -\gamma_1 & \dots & 0 \\ \dots & \dots & \dots & \dots \\ \beta_n & 0 & \dots & -\gamma_n \end{bmatrix} \begin{bmatrix} P_0 \\ P_1 \\ \dots \\ P_n \end{bmatrix} = \begin{bmatrix} \alpha \\ 0 \\ \dots \\ 0 \end{bmatrix}. \tag{12}$$

Equilibrium abundances of different isoforms can be obtained by solving the above system of equations.

In this study, we focused on two types of gene product modification processes, editing-type and splicing-type, which are exemplified by RNA editing and splicing, respectively. A variant of the above model is applied to each of the two types. For editing-type modification, we considered a simple case with two isoforms: the unmodified isoform $I_0$ and the modified isoform $I_1$. Equilibrium isoform abundances were calculated simply using Eq. (9). When deriving model predictions, we had $\gamma_0 = 1$ and $\gamma_1 = 1$, unless stated otherwise. For splicing-type modification, we considered a model with three isoforms: the unmodified isoform $I_0$ and two modified isoforms, $I_1$ and $I_2$. Equilibrium isoform abundances were calculated by solving Eq. (12) with $n = 2$. When deriving model predictions, we had $\gamma_0 = 0$ and $\gamma_1 = 1$, unless stated otherwise.

The modeling framework also extends to multi-step modification, where a modified isoform can be further modified into a different one. Let us consider a scenario where a modified isoform $I_1$ is modified into a different isoform $I_2$. The equilibrium is reached when

$$
\begin{cases}
\frac{dP_0}{dt} = \alpha - \beta_{0\to1}P_0 - \gamma_0 P_0 = 0 \\
\frac{dP_1}{dt} = \beta_{0\to1}P_0 - \beta_{1\to2}P_1 - \gamma_1 P_1 = 0 \\
\frac{dP_2}{dt} = \beta_{1\to2}P_1 - \gamma_2 P_2 = 0.
\end{cases}
\tag{13}
$$

Solving the above system of equations gives

$$
\begin{cases}
P_0 = \frac{\alpha}{\beta_{0\to1}+\gamma_0} \\
P_1 = \frac{\alpha\beta_{0\to1}}{(\beta_{0\to1}+\gamma_0)(\beta_{1\to2}+\gamma_1)} \\
P_2 = \frac{\alpha\beta_{0\to1}\beta_{1\to2}}{(\beta_{0\to1}+\gamma_0)(\beta_{1\to2}+\gamma_1)\gamma_2}.
\end{cases}
\tag{14}
$$

Similarly, if there is a series of $n$ modified isoforms where $I_i$ is produced by modifying $I_{i-1}$:

$$
\begin{cases}
\frac{dP_0}{dt} = \alpha - \beta_{0\to1}P_0 - \gamma_0 P_0 = 0 \\
\frac{dP_1}{dt} = \beta_{0\to1}P_0 - \beta_{1\to2}P_1 - \gamma_1 P_1 = 0 \\
\dots, \\
\frac{dP_{n-1}}{dt} = \beta_{n-2\to n-1}P_{n-2} - \beta_{n-1\to n}P_{n-1} - \gamma_{n-1} P_{n-1} = 0 \\
\frac{dP_n}{dt} = \beta_{n-1\to n}P_{n-1} - \gamma_n P_n = 0.
\end{cases}
\tag{15}
$$

The above system of equations can be rearranged and written in a matrix ($\mathbf{A}x = \mathbf{b}$) form:

$$
\begin{bmatrix}
\beta_{0\to1}+\gamma_0 & 0 & \dots & 0 & 0 & 0 \\
\beta_{0\to1} & -\beta_{1\to2}-\gamma_1 & \dots & 0 & 0 & 0 \\
\dots & \dots & \dots & \dots & \dots & \dots \\
0 & 0 & \dots & \beta_{n-2\to n-1} & -\beta_{n-1\to n}-\gamma_{n-1} & 0 \\
0 & 0 & \dots & 0 & \beta_{n-1\to n} & -\gamma_n
\end{bmatrix}
$$
$$
\begin{bmatrix} P_0 \\ P_1 \\ \dots \\ P_{n-1} \\ P_n \end{bmatrix}
=
\begin{bmatrix} \alpha \\ 0 \\ \dots \\ 0 \\ 0 \end{bmatrix}.
\tag{16}
$$

It is worth noting that the above model can be applied when it is the number of modification events within the same RNA or protein molecule but not the exact locations of the modifications that are of interest. In such a case, $n$ represents the total number of sites in the RNA or protein molecule that can potentially be modified, and $I_i$ represents isoforms where $i$ of the $n$ potential sites are modified. If the per-site modification rate is constant regardless of the location of the potential modification site or modification states of other sites, such that for each $0\leq i\leq n-1$ there is $\beta_{i\to i+1} = (n-i)\beta$, Eqs. (15) and (16) can be written as

$$
\begin{cases}
\frac{dP_0}{dt} = \alpha - n\beta P_0 - \gamma_0 P_0 = 0 \\
\frac{dP_1}{dt} = n\beta P_0 - (n-1)\beta P_1 - \gamma_1 P_1 = 0 \\
\dots, \\
\frac{dP_{n-1}}{dt} = 2\beta P_{n-2} - \beta P_{n-1} - \gamma_{n-1} P_{n-1} = 0 \\
\frac{dP_n}{dt} = \beta P_{n-1} - \gamma_n P_n = 0
\end{cases}
\tag{17}
$$

and

$$
\begin{bmatrix}
n\beta+\gamma_0 & 0 & \dots & 0 & 0 & 0 \\
n\beta & -(n-1)\beta-\gamma_1 & \dots & 0 & 0 & 0 \\
\dots & \dots & \dots & \dots & \dots & \dots \\
0 & 0 & \dots & 2\beta & -\beta-\gamma_{n-1} & 0 \\
0 & 0 & \dots & 0 & \beta & -\gamma_n
\end{bmatrix}
$$
$$
\begin{bmatrix} P_0 \\ P_1 \\ \dots \\ P_{n-1} \\ P_n \end{bmatrix}
=
\begin{bmatrix} \alpha \\ 0 \\ \dots \\ 0 \\ 0 \end{bmatrix}.
\tag{18}
$$

In the most general form of the model where every isoform, $I_i$, can be converted to another isoform, $I_j$ (where $i \neq j$), at per-molecule rate $\beta_{i,j}$ ($\beta_{i,j} = 0$ if $i = j$), Eq. (12) will be written as

$$
\begin{bmatrix}
\sum_{i=1}^{n}\beta_{0,i}+\gamma_0 & 0 & \dots & 0 \\
\beta_{0,1} & -\sum_{i=0}^{n}\beta_{1,i}-\gamma_1 & \dots & \beta_{n,1} \\
\dots & \dots & \dots & \dots \\
\beta_{0,n} & \beta_{1,n} & \dots & -\sum_{i=0}^{n}\beta_{n,i}-\gamma_n
\end{bmatrix}
$$
$$
\begin{bmatrix} P_0 \\ P_1 \\ \dots \\ P_n \end{bmatrix}
=
\begin{bmatrix} \alpha \\ 0 \\ \dots \\ 0 \end{bmatrix}.
\tag{19}
$$

The above model can also be modified to model alternative outcomes of the same gene's transcription, such as the use of alternative promoters. In such a case, equilibrium abundance of the $i$th isoform is obtained when

$$
\frac{dP_i}{dt} = \alpha_i - \gamma_i P_i = 0,
\tag{20}
$$

where $\alpha_i$ is the rate at which the $n$th isoform is produced. Solving the equation gives simply $P_i = \frac{\alpha_i}{\gamma_i}$. If there are two alternative isoforms ($I_0$ and $I_1$), and the total rate of transcription $\alpha$ is constant, equilibrium is reached when

$$
\begin{cases}
\frac{dP_0}{dt} = (1-E)\alpha - \gamma_0 P_0 = 0 \\
\frac{dP_1}{dt} = E\alpha - \gamma_1 P_1 = 0
\end{cases},
\tag{21}
$$

where $E$ is the possibility that $I_1$ is transcribed given that transcription happens and can be interpreted as an error rate if $I_0$ is the functional isoform and $I_1$ is not. The solution is then

$$\begin{cases} P_0 = \frac{(1-E)\alpha}{\gamma_0} \\ P_1 = \frac{E\alpha}{\gamma_1} \end{cases}. \tag{22}$$

While such diversity can indeed be modeled under our framework, it is not a focus of this study and will not be discussed further in this paper.

## Genetic architecture of modification rate

For a given modified isoform, the corresponding $\beta$ parameter is determined together by $lcis$-acting loci and a $trans$-genotypic value, $Q$. The $trans$-genotypic value $Q$ characterizes the overall activity of the enzyme or molecular machinery that carries out the modification process, and is a product of its expression level and per-molecule activity. The binding affinity between the enzyme and its substrate is dependent on the $cis$-loci, which are genomic loci encoding regions adjacent to (though not necessarily immediately adjacent to) the site subject to modification.

We assumed that each $cis$-locus can have either an effector allele that facilitates binding between the modification enzyme and its substrate, or a null allele that does not facilitate binding. The total effect of the $cis$-genotype on $\beta$ is determined by a normalized genotypic value $\hat{v}$, which is calculated as

$$\hat{v} = \frac{v}{v_{max}}, \tag{23}$$

where $v$ is the sum of all effector alleles' effect, and $v_{max}$ is the greatest possible value of $v$, when there are no null alleles. We assume that the $cis$-loci's effect is additive and all $cis$-loci have equal effect, so there $v$ is equal to the total number of effector alleles, and $v_{max}$ is equal to the number of $cis$-loci, $l$.

The relationship between $\beta$ and underlying parameters is given by

$$\beta = Q(C\hat{v} + \epsilon) \text{ where } C > 0, \epsilon \geq 0. \tag{24}$$

Here, $\epsilon$ is the rate of nonspecific modification that takes place independent of the $cis$-genotype, and $C$ reflects global structural features of an RNA or protein molecule that affect binding affinity between the enzyme and the substrate.

For splicing-type modification, we assumed that $\beta_1$ and $\beta_2$ are affected by the same set of $cis$-loci. We also assumed the two alleles that each $cis$-locus could potentially have are both effector alleles. One of them only facilitates the production of $I_1$, whereas the other only facilitates the production of $I_2$; under this model, the same genotype's effects on $I_1$ and $I_2$ are inversely correlated. For convenience, we defined the normalized $cis$-genotypic value based on the genotype's effect on $\beta_1$. The $\beta$ parameters are thus given by

$$\beta_1 = Q(C\hat{v} + \epsilon) \tag{25}$$

and

$$\beta_2 = Q(C(1 - \hat{v}) + \epsilon). \tag{26}$$

For editing-type modification, we had $C = 1$, $Q = 1$, and $epsilon = 0$ when deriving model predictions, unless specified otherwise. For splicing-type, we had $C = 1$, $Q = 100$, and $epsilon = 0$, unless specified otherwise.

The mutational spectrum of each $cis$-locus is characterized by two per-locus mutation rates: $\mu_{01}$, the rate of mutations from the null allele to the effector allele, and $\mu_{10}$, the rate of mutations in the opposite direction. The difference between $\mu_{01}$ and $\mu_{10}$ reflects a difference in the two allele's sequence spaces and/or rate of different types of nucleotide changes (e.g., transition/transversion bias or AT-bias). In the case of splicing-type modification, $\mu_{01}$ and $\mu_{10}$ are simply replaced by mutation rates between two effector alleles. For simplicity, we assumed that all $cis$-loci have the same mutational spectrum in this study. In this study, we had the total mutation rate per locus $\mu_{01} + \mu_{10} = 2 \times 10^{-9}$, unless stated otherwise.

The $trans$-genotypic value $Q$ is modeled as a continuous trait in this study, and the effect of each mutation on $\ln Q$ was sampled from a normal distribution $\mathcal{N}(0, S_Q)$. In this study, we assumed that $trans$-mutations do not affect binding specificity; that is, they recognize the same $cis$-motifs. In some modification systems, however, the $cis$-$trans$ interaction is strongly sequence-specific such that mutations could make $trans$-factor's interact with a different set of targets. Such examples include RNA editing in kinetoplasts, which involves guide RNAs (Hajduk and Ochsenreiter, 2010), and RNA processing by PRR proteins in plants (Shikanai and Fujii, 2013).

## Selection of isoform abundance

We first considered a scenario where each isoform contributes to fitness independently, in which case the fitness is given by

$$\omega = \prod_{i=0}^{n} \omega_i, \tag{27}$$

where $\omega_i$ is fitness with respect to $P_i$.

We considered two scenarios where an isoform's abundance is subject to selection: a scenario where the isoform is functional and a scenario where it is not functional but deleterious. For a functional isoform, $I_i$, the relationship between its abundance, $P_i$, and fitness is characterized by a Gaussian fitness function:

$$\omega_i = \exp\left(-\frac{\ln P_i - \ln \tilde{P}_i}{2\sigma_i^2}\right), \tag{28}$$

where $\sigma_i^2$, the width of the fitness function, and describes the strength of selection.

If $I_i$ is deleterious, fitness with respect to its abundance $P_i$ is given by

$$\omega_i = \exp(-\lambda_i P_i), \tag{29}$$

where $\lambda_i > 0$ is a parameter characterizing the strength of selection. When $\lambda_i = 0$, there is $\omega_i = 1$, which corresponds to the case that $P_i$ is not under selection. In this study, we had $\sigma = 10$ for every functional

isoform and $\lambda = 10^{-3}$ for every deleterious isoform, unless specified otherwise.

For editing-type modifications, we mainly focused on a scenario where the modification is deleterious: here, $I_0$ is functional while $I_1$ is toxic. The fitness component with respect to $P_0$ is calculated by Eq. (28), whereas fitness with respect to $P_1$ is calculated by Eq. (29). For splicing-type modifications, fitness is determined only by $P_1$ and $P_2$, not $P_0$. One of the modified isoforms, $I_1$, is the functional, and its abundance $P_1$ is under stabilizing selection; the fitness component with respect to $P_1$ is thus computed using Eq. (28). The other modified isoform, $I_2$, in contrast, is not functional but toxic, and the corresponding fitness component is computed using Eq. (29). With $I_2$ representing mis-processed isoform(s), we also assumed that $\gamma_2$ is greater than $\gamma_1$ to recapitulate quality-control mechanisms that act to eliminate mis-processed isoforms (Frischmeyer and Dietz, 1999; Kurosaki and Maquat, 2016; Kurosaki et al, 2019); specifically, we examined scenarios of $\gamma_1 = 1$ while $\gamma_2$ is equal to 20, 50, or 100.

## Distribution of *cis*-genotypic value

When the number of *cis*-loci underlying a modification event is reasonably small, the evolution of genotypic value $v$ (and thus $\hat{v}$) can be approximated by a sequential-fixation (strong-selection-weak-mutation) model (McCandlish and Stoltzfus, 2014). Then, assuming that other parameters that affect modification are constant, the evolution of $v$ (and $\hat{v}$) can be modeled as a Markov process with a constant transition matrix. A time step in this Markov process can be a generation or any arbitrary time interval as long as the the probability that more than one mutations arise in the population is very low ($2N_e\mu$ <0.01) such that the sequential-fixation model is an appropriate approximation (Lynch, 2020). Using this approach, the distribution of the *cis*-genotypic value $v$ given the starting state after a given amount of time can be derived.

Let us consider a simple scenario where the effector allele at every *cis*-locus has an effect size of 1 where $v$ is equal to the number of effector alleles and $v_{max} = l$. In a diploid population, the probability that $v$ becomes $v + 1$ via substitution in a time step given the present genotypic value $v$ is

$$\Pr(v \to v + 1) = 2(l - v)N_e u_{01} f_{v \to v+1}, \qquad (30)$$

where $N_e$ is the effective population size and $f_{v \to v+1}$ is the fixation probability given ancestral and mutant phenotypes.

Similarly, the probability of becoming $v - 1$ via a substitution is

$$\Pr(v \to v - 1) = 2vN_e u_{10} f_{v \to v-1}. \qquad (31)$$

The probability that $v$ does not change is simply

$$\Pr(v \to v) = 1 - \Pr(v \to v + 1) - \Pr(v \to v - 1). \qquad (32)$$

The fixation probability is obtained using Kimura's method (Kimura, 1962):

$$\Pr(\text{fixation}|N_e, s) = \frac{1 - \exp(-2s)}{1 - \exp(-4N_e s)},$$

where $s = \frac{\omega_M}{\omega_A} - 1$ is the coefficient of selection ($\omega_M$ and $\omega_A$ represent mutant and ancestral fitness, respectively).

Given the probability distribution of $v$ at a time $t$, $\mathbf{v}_t$, the distribution at $t + 1$ is

$$\mathbf{v}_{t+1} = \mathbf{v}_t \mathbf{T}, \qquad (33)$$

where $\mathbf{v}_t$ and $\mathbf{v}_{t+1}$ are row vectors of length $l + 1$, with each element represents the probability of a possible value of $v$. The transition matrix $\mathbf{T}$ is a $l + 1 \times l + 1$ matrix where $\mathbf{T}[i + 1, j + 1] = \Pr(i \to j)$. The probability $\Pr(i \to j)$ is calculated following Eqs. (30) and (31) if $0 \le i \le l$, $0 \le j \le l$, and $|i - j| \le 1$; otherwise, $\Pr(i \to j) = 0$. In this study, we used $\mathbf{v}_0 \mathbf{T}^{1e8}$ to represent an equilibrium distribution. For editing-type modification, we had the first element of $\mathbf{v}_0$ equal to 1 (i.e., starting from the genotype that has the least effect on modification), whereas for splicing-type modification, we had the last element of $\mathbf{v}_0$ equal to 1 (i.e., starting from the genotype that maximizes the production of $I_1$ and minimized the production of $I_2$).

If different *cis*-loci have different effect sizes, there will be up to $\binom{l}{2}$ possible values of $v$. In the extreme case where all loci have different effect sizes, and the mutation rate depends both on the locus and the ancestral allele, the transition probability from a given genotype to a given neighbor genotype (one mutation removed from the ancestral genotype) is simply the product of the local mutation rate and the fixation probability. In this study, we focus mainly on the simple scenario where all the *cis*-loci have equal effect size and mutation rates, although the modeling framework can be easily extended to more general cases.

In this manuscript, we mainly present results after evolution for $10^8$ generations to represent long-term evolution instead of stationary distributions reached as $t \to \infty$, as a time interval of $10^8$ generations is rather long and can readily be considered at macroevolutionary timescale.

## Simulating *cis*-*trans* coevolution

To investigate coevolutionary dynamics between the *cis*-loci and the *trans*-genotypic value $Q$ when many genes or sites are subject to modification, we conducted simulations of evolution where *cis*-loci and $Q$ are both affected by mutations.

Each lineage we simulated was divided into a number of time steps, with the number of time steps proportional to the branch length. If the only loci that could undergo evolutionary changes in a time interval are the *cis*-loci, the probability distribution of a given modification event's *cis*-genotypic value $v$ at the end of the time interval is simply

$$\mathbf{v}_t = \mathbf{v}_0 \mathbf{T}^t \qquad (34)$$

where $v_0$ is the starting distribution and $t$ is the number of time steps the time interval consists of. If the simulation starts from a pre-designated value of $v$, the corresponding element of $\mathbf{v}_0$ will be 1 while other elements are equal to 0.

Before simulating evolution for a lineage, we first determined $m$, the total number of mutations that affect $Q$ to occur during evolution by sampling $m$ from a Poisson distribution with mean equal to $2N_e U_Q L$, where $L$ is the branch length (number of time steps) and $U_Q$ is the rate of mutations that affect $Q$. Then we randomly picked $m$ time steps, at each of which a mutation affecting $Q$ would occur. If $m > L$ (which has very low probability

given parameter values considered, and did not happen in our simulations), this value of $m$ will not be used for simulations. The effect of each mutation on $\ln Q$ was then sampled from $\mathcal{N}(0, S_Q)$. Change in the distribution $v$ during the interval between two mutations that affect $Q$ obtained using Eq. (34), with $t$ being the number of time steps between two mutations. Before examining the fitness effect of a mutation that affects $Q$, a value of $v$ was first sampled from its distribution, which, together with the mutation's effect on $Q$, will determine the fixation probability. If the mutation is fixed, the transition matrix will be re-calculated with the mutant $Q$, and the mutant $Q$ will be the new $Q$ to begin with when the next mutation is examined. When products of multiple genes are subject to modification, fitness effect of each mutation affecting $Q$ is determined collectively by its effect on all modification events; when such a mutation is fixed, all gene's transition matrices will be altered. For simplicity, we assumed that different modification events' cis-loci are not shared and evolve independently.

We considered a scenario where the modification machinery has both beneficial and detrimental effects on fitness at the same time. Under this model, there are a set of genes subject to deleterious editing-type modifications, where the unmodified isoform is functional and the modified isoform is deleterious. At the same time, $Q$ contributes to a fitness component $\omega_Q$ that is independent of these modification events. In our simulations, $Q$ was under stabilizing selection, and $\omega_Q$ is given by

$$\omega_Q = \exp\left(-\frac{\ln Q - \ln \tilde{Q}}{2\sigma_Q^2}\right), \tag{35}$$

where $\tilde{Q}$ is the optimal value of $Q$ and $\sigma_Q$ is the fitness function's width. In this case, if there are $n$ genes subject to modification, the overall fitness is given by

$$\omega = \omega_Q \prod_{i=0}^{n} \omega_i, \tag{36}$$

where $\omega_i$ is fitness with respect to the $i$th gene's isoform abundances.

Values of $N_e$ used in the simulations include $10^2$, $10^{2.5}$, $10^3$, $10^{3.5}$, $10^4$, $10^{4.5}$, and $10^5$. In each simulation, we considered 100 genes that are subject to deleterious modifications. For simplicity, we had all modification events have equal $l$, and considered scenarios of $l = 2$, $l = 5$, and $l = 10$, where the initial value of $v$ was 1, 2, and 5, respectively. Regarding selection on $Q$, we considered two scenarios: a scenario of strong selection ($\sigma_Q = 2$) and a scenario of relatively weak selection ($\sigma_Q = 20$). In all simulations, we had $\tilde{Q} = 2$, $U_Q = 10^{-8}$ and $S_Q = 0.1$. We also had $\alpha = 1$, $\gamma_0 = 1$, $\gamma_1 = 1$, $C = 1$, $\sigma = 10$, $\lambda = 10^{-3}$, and $\epsilon = 10^{-3}$ for all genes in all simulations. Starting value of $Q$ was equal to its optimum for all simulations. After the simulations, we quantified the degree to which the modifications are shared among lineages. For each gene, we calculated the fraction of lineages where $P_1 > 0.005$. The median of all genes is then used to represent how likely a modification event is shared given the evolutionary parameters ($l$, $N_e$, and strength of selection). We examined how this value varied depending on divergence time by performing the simulation with different times of duration, including $2 \times 10^7$, $4 \times 10^7$, $6 \times 10^7$, $8 \times 10^7$, and $10^8$ time steps. For each combination of parameter values, we simulated 50 independent lineages.

The above procedure can also be used to simulate the coevolution of the cis-loci and other parameters, such as $\alpha$, $C$ or $\epsilon$, in which case mutations affecting $Q$ in the above procedure will be replaced by mutations affecting the parameter of interest.

## Simulation along the coleoid tree

We simulated evolution of editing levels at 20,000 editing sites along a phylogenetic tree of four coleoid species: the common octopus (*Octopus vulgaris*), the bimac (*O. bimaculoides*), the squid (*Doryteuthis pealeii*), and the cuttlefish (*Sepia oficianalis*). The coleoids have high A-to-I RNA editing activity in their neural tissues, whereas extant non-coleoid cephalopods and non-cephalopod mollusks do not (Alon et al, 2015; Liscovitch-Brauer et al, 2017). Branch lengths of the phylogenetic tree are based on divergence times described in ref. Liscovitch-Brauer et al, 2017, with mid point the reported range used for our simulations. Divergence time of the octopus and the bimac, which are very closely related, was set to be 5 million years. We assumed each time step in the simulation corresponds to a year, so the number of time steps a branch corresponds to is equal to branch length in terms of years. We started the simulation from the most recent common ancestor of four coleoids, and the value of $v$ of each editing site at this ancestral node was sampled randomly from the corresponding genotypic space. We assumed that $Q$ is under strong stabilizing selection mediated by functions independent of the focal editing events such that $Q$ remained constant in the simulation. We had $Q = 1$ for this simulation. The distribution of $v$ at the end of each branch was obtained using Eq. (34) with time of evolution equal to branch length; a value of $v$ was then sampled from the distribution to represent the state at the end of this branch and the starting state of its descendent branches (if any). Some gene-specific parameters were sampled from pre-specified distributions. The rate at which $I_0$ is expressed, $\alpha$, was sampled from a log-normal distribution; that is, $\ln \alpha$ was sampled from $\mathcal{N}(0, 1)$. The number of cis-loci, $l$, was sampled uniformly from $(0, 1, \ldots, 10)$. The $C$ parameter was sampled from a exponential distribution with mean equal to 0.1. All genes had $\gamma_0 = 1$, $\gamma_1 = 1$, $\sigma = 10$, $\lambda = 10^{-3}$, and $\epsilon = 10^{-4}$. Because $\epsilon > 0$, all editing levels were positive. Thus, after the simulation, we log-transformed all editing levels and computed Euclidean distances between each pair of species using log-transformed editing levels ($\ln(f)$). We then built a neighbor-joining (NJ) tree based on these distances using the *nj* function of R package *ape*, and asked this NJ tree to recapitulate the phylogenetic relationship of the four coleoid species; specifically, we examined whether (1) the two *Octopus* species fall in one clade while the squid and the cuttlefish fall in another, and (2) whether distance between the two octopuses is closer than that between the squid and the cuttlefish.

## Modeling computer scripts

GitHub (https://github.com/applied-phylo-lab/gene_product_diversity).

# Data availability

The datasets and computer code produced in this study are available in the following databases:

The source data of this paper are collected in the following database record: biostudies:S-SCDT-10_1038-S44320-025-00095-4.

## Peer review information

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

## Acknowledgements

The authors thank Mark Kim, and members of the Pennell, Edge, and Mooney labs for their thoughtful comments on parts of this study. We acknowledge support from the Natural Sciences and Engineering Research Council of Canada (FN 492860, to AFP), the Jean D'Alembert Foundation (France 2030 program ANR-11-IDEX-0003, to AFP), and the National Institute of General Medical Sciences (R35GM151348, to MP).

## Author contributions

**Daohan Jiang**: Conceptualization; Formal analysis; Investigation; Visualization; Methodology; Writing—original draft; Writing—review and editing. **Nevraj Kejiou**: Conceptualization; Writing—review and editing. **Yi Qiu**: Conceptualization; Writing—review and editing. **Alexander F Palazzo**: Conceptualization; Investigation; Writing—original draft; Writing—review and editing. **Matt Pennell**: Conceptualization; Funding acquisition; Methodology; Writing—original draft; Writing—review and editing.

Source data underlying figure panels in this paper may have individual authorship assigned. Where available, figure panel/source data authorship is listed in the following database record: biostudies:S-SCDT-10_1038-S44320-025-00095-4.

## Disclosure and competing interests statement

The authors declare no competing interests.

# Expanded View Figures

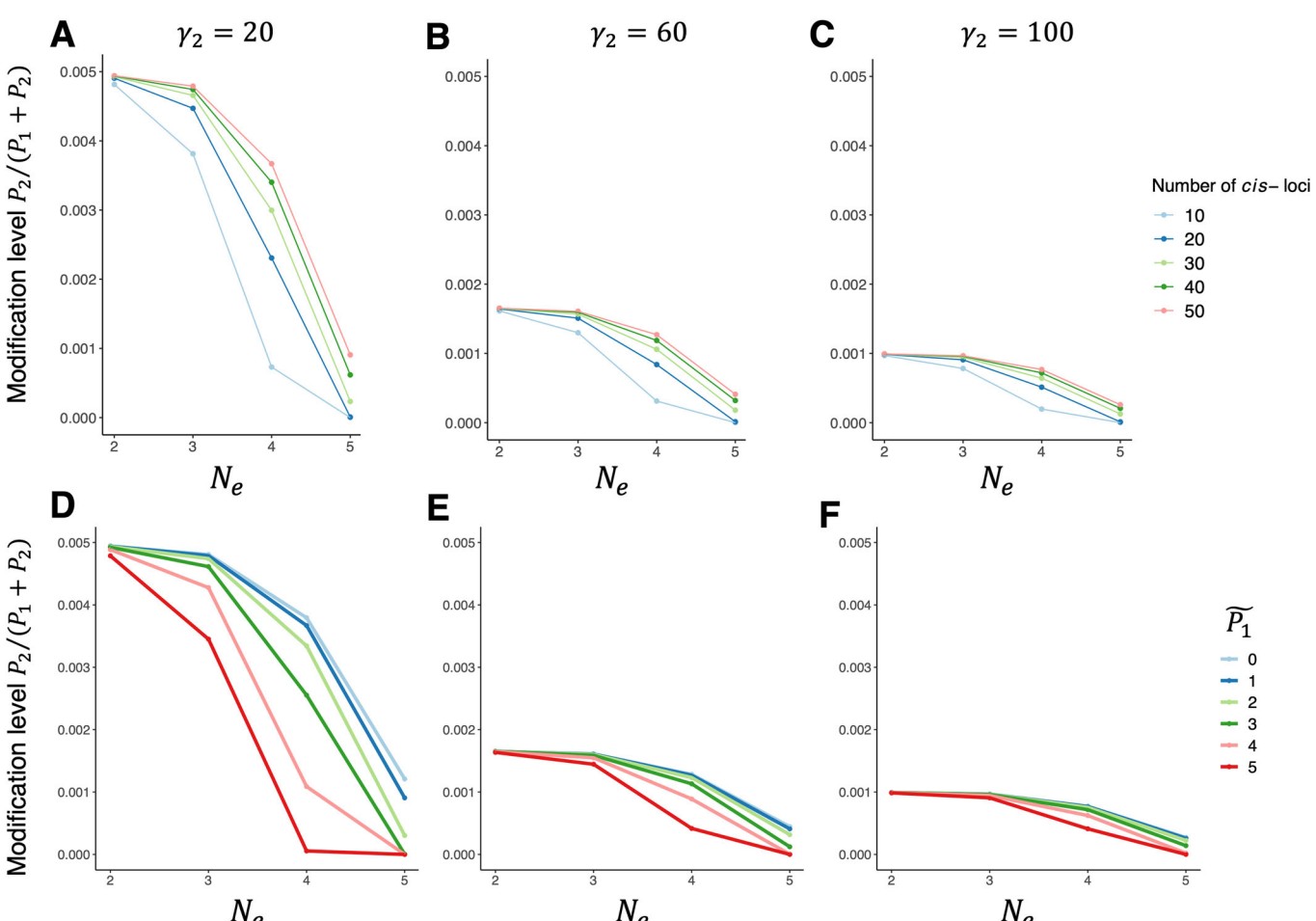

**Figure EV1. Mean modification level varies with population genetic environment and genetic architecture.**

Scaling between mean modification level of splicing-type modification and effective population size $N_e$ (shown in log10 scale). (**A–C**) Response of mean modification level to $N_e$ under different combinations of *cis*-loci number ($l$) and decay rates of the dysfunctional isoform ($\gamma_2$), with optimal expression level $\tilde{P}_1 = \exp(1)$ ($\ln \tilde{P}_1 = 1$). (**D–F**) Response of mean modification level to $N_e$ under different $\tilde{P}_1 = \frac{a}{\gamma_1}$ and $\gamma_2$, with $l = 50$. All results are derived with initial *cis*-genotypic value $v_0 = l$, with $T = 10^8$ time steps, $\mu_{01} = \mu_{10} = 10^{-8}$, $Q = 100$, $\gamma_0 = 0$, $\gamma_1 = 1$, and $\tilde{P}_1 = a/\gamma_1$.

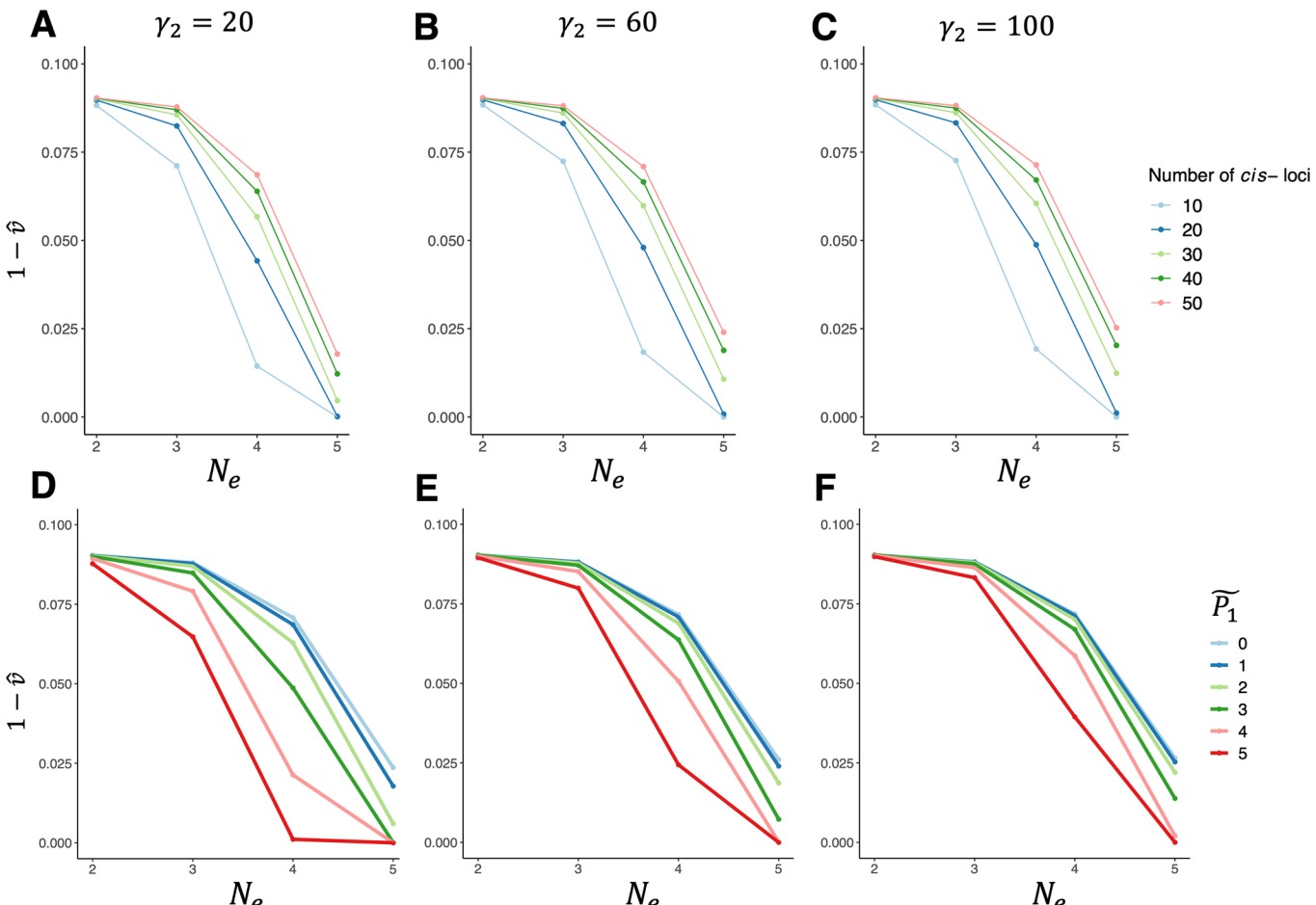

**Figure EV2. Cis-genotypic value varies with population genetic environment and genetic architecture.**

Scaling between normalized mean *cis*-genotypic value of splicing-type modification and $N_e$ (shown in log10 scale). Represented by the Y-axes is $1 - \hat{v}$, which reflects the degree to which *cis*-genotype favors production of the dysfunction and toxic isoform $I_2$. (A–C) Response of $1 - \hat{v}$ to $N_e$ under different combinations of $l$ and $\gamma_2$, with optimal expression level $\bar{P}_1 = \exp(1)$ ($\ln \bar{P}_1 = 1$). (D–F) Response of $1 - \hat{v}$ to $N_e$ under different $\bar{P}_1$ and $\gamma_2$, with $l = 50$. All results are derived with initial *cis*-genotypic value $v_0 = l$, time of evolution $T = 10^8$ time steps, and $\mu_{O1} = \mu_{1O} = 10^{-8}$, $Q = 100$, $\gamma_0 = 0$, $\gamma_1 = 1$, and $\bar{P}_1 = a/\gamma_1$.

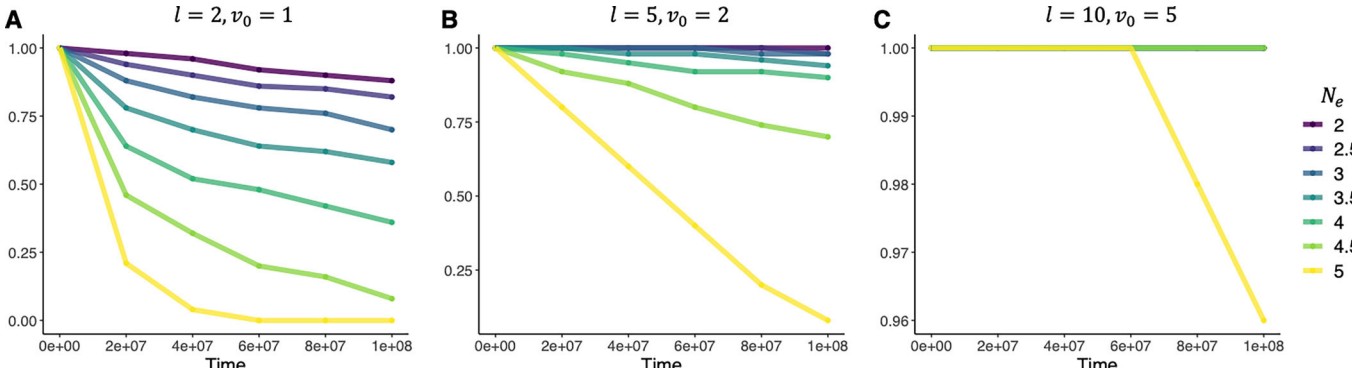

**Figure EV3.  Conservation of modification events as a function of time since divergence.**

(A) $l = 2$, $v_0 = 1$. (B) $l = 5$, $v_0 = 2$. (C) $l = 10$, $v_0 = 5$. Y-axes represent among-gene median of proportion of lineages (species) that share a modification event when selection on Q is weak ($\sigma_Q = 20$). When two curves in the same panel completely overlap, the one with the largest corresponding $N_e$ is shown.

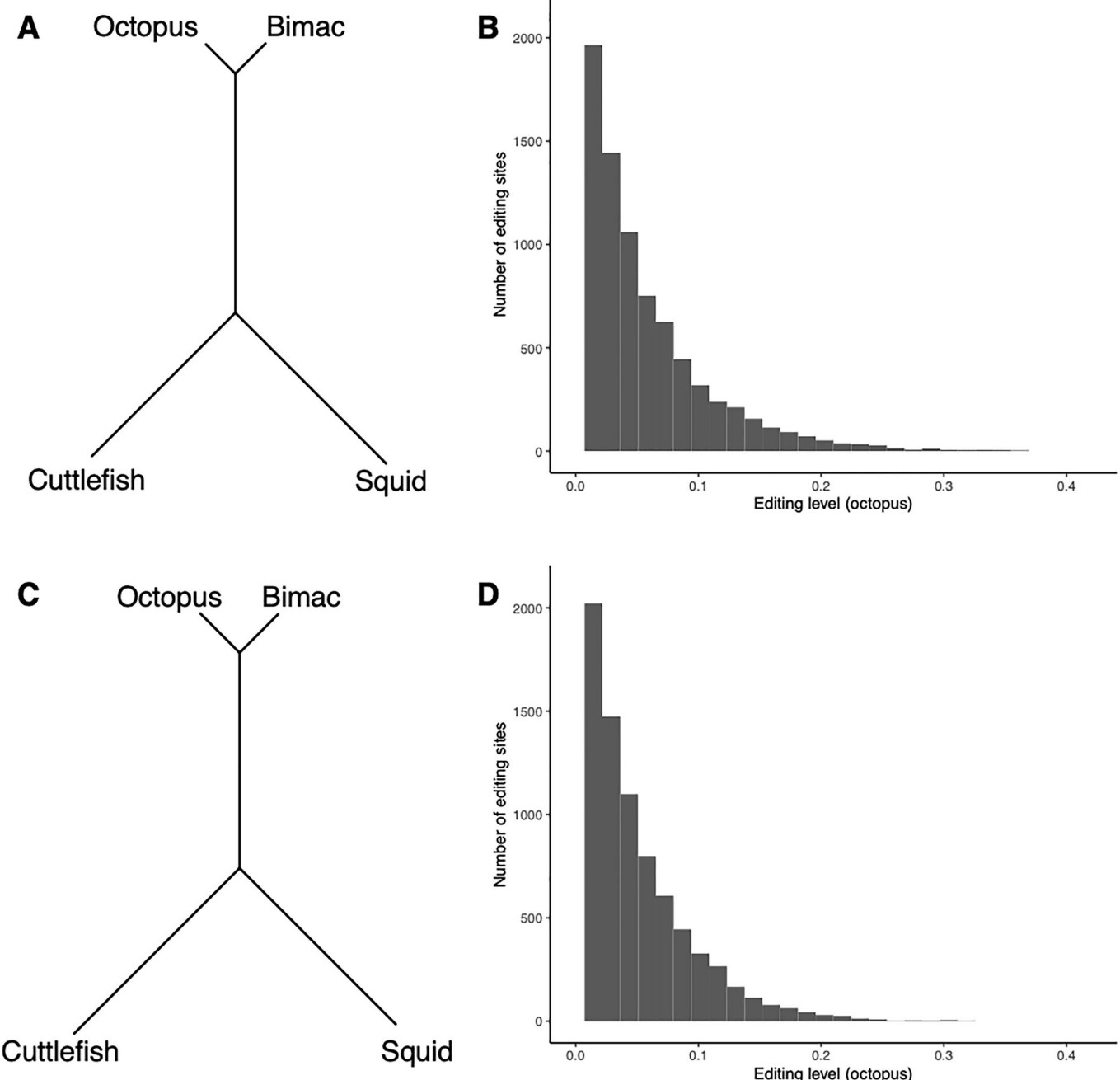

**Figure EV4.  Simulations of A-to-I RNA editing along the coleoid phylogeny.**

(A) Neighbor-joining tree of four coleoid species based on simulated neutral editing levels. (B) Distribution of neutral editing levels in the octopus. (C) Neighbor-joining tree of four coleoid species based on simulated deleterious editing levels. (D) Distribution of deleterious editing levels in the octopus.

