## [Peer Review File · Molecular Systems Biology]

Constraints on the optimization of gene product diversity

Daohan Jiang, Nevraj Kejiou, Yi Qiu, Alexander Palazzo, and Matt Pennell

Corresponding author(s): Matt Pennell (mpennell@cornell.edu) , Alexander Palazzo (alex.palazzo@utoronto.ca)

Review Timeline:

Submission Date:	8th Sep 24
Editorial Decision:	28th Oct 24
Revision Received:	17th Nov 24
Editorial Decision:	25th Feb 25
Revision Received:	27th Feb 25
Accepted:	13th Mar 25

Editor: Poonam Bheda

Transaction Report:

28th Oct 2024

Manuscript Number: MSB-2024-12620

Title: Genetic and selective constraints on the optimization of gene product diversity

Dear Dr. Pennell,

Thank you for the submission of your manuscript to Molecular Systems Biology. We have now received feedback from the three reviewers who agreed to evaluate your manuscript. As you will see from the reports below, the referees acknowledge the interest of the study and are overall supporting publication of your work pending appropriate revisions.

I think that the recommendations of the reviewers are rather clear and I therefore do not see the need to repeat the comments listed below. One of the more fundamental points raised refers to the need to validate the predictions made by your model on alternative splicing as suggested by Reviewer 1 and also reflecting our initial editorial impression of the paper. All other issues raised would need to be satisfactorily addressed. Please let me know in case you would like to discuss in further detail any of the issues raised, I would be happy to schedule a call.

We require:

1) A .docx formatted version of the manuscript text (including legends for main figures, EV figures and tables). Please make sure that the changes are highlighted to be clearly visible. Alternatively you may choose to submit your manuscript as a LaTeX file.

4) A .docx formatted letter INCLUDING the reviewers' reports and your detailed point-by-point responses to their comments. As part of the EMBO Press transparent editorial process, the point-by-point response is part of the Peer Review File (PRF), which will be published alongside your paper.

5) A complete author checklist, which you can download from our author guidelines (<https://www.embopress.org/page/journal/17574684/authorguide#submissionofrevisions>). Please insert information in the checklist that is also reflected in the manuscript. The completed author checklist will also be part of the PRF.

6) Please note that all corresponding authors are required to supply an ORCID ID for their name upon submission of a revised manuscript.

7) It is mandatory to include a 'Data Availability' section after the Materials and Methods. Before submitting your revision, primary datasets produced in this study need to be deposited in an appropriate public database, and the accession numbers and database listed under 'Data Availability'. Please remember to provide a reviewer password if the datasets are not yet public (see <https://www.embopress.org/page/journal/17574684/authorguide#dataavailability>).

In case you have no data that requires deposition in a public database, please state so in this section. Note that the Data Availability Section is restricted to new primary data that are part of this study. This study includes no data deposited in external repositories.

8) All Materials and Methods need to be described in the main text using our 'Structured Methods' format, which is required for all research articles. According to this format, the Methods section includes a Reagents and Tools Table (listing key reagents, experimental models, software and relevant equipment and including their sources and relevant identifiers) followed by a Methods and Protocols section describing the methods using a step-by-step protocol format. The aim is to facilitate adoption of the methodologies across labs. Please upload the Reagents and Tools table as a separate document when submitting your revised manuscript. More information on how to adhere to this format as well as a downloadable template (.docx) for the Reagents and Tools Table can be found in our author guidelines: <https://www.embopress.org/page/journal/17444292/authorguide#structuredmethods>

An example of a Method paper with Structured Methods can be found here:
<https://www.embopress.org/doi/10.15252/msb.20178071>.

9) For data quantification: please specify the name of the statistical test used to generate error bars and P values, the number (n) of independent experiments (specify technical or biological replicates) underlying each data point and the test used to calculate p-values in each figure legend. The figure legends should contain a basic description of n, P and the test applied. Graphs must include a description of the bars and the error bars (s.d., s.e.m.). Please provide exact p values.

10) Our journal encourages inclusion of *data citations in the reference list* to directly cite datasets that were re-used and obtained from public databases. Data citations in the article text are distinct from normal bibliographical citations and should directly link to the database records from which the data can be accessed. In the main text, data citations are formatted as follows: "Data ref: Smith et al, 2001" or "Data ref: NCBI Sequence Read Archive PRJNA342805, 2017". In the Reference list, data citations must be labeled with "[DATASET]". A data reference must provide the database name, accession number/identifiers and a resolvable link to the landing page from which the data can be accessed at the end of the reference. Further instructions are available at .

11) We replaced Supplementary Information with Expanded View (EV) Figures and Tables that are collapsible/expandable online. A maximum of 5 EV Figures can be typeset. EV Figures should be cited as 'Figure EV1, Figure EV2' etc... in the text and their respective legends should be included in the main text after the legends of regular figures.

<https://www.embopress.org/page/journal/17574684/authorguide#expandedview>

13) Author contributions: CRediT has replaced the traditional author contributions section because it offers a systematic machine readable author contributions format that allows for more effective research assessment. Please remove the Authors Contributions from the manuscript and use the free text boxes beneath each contributing author's name in our system to add specific details on the author's contribution. More information is available in our guide to authors.

14) Disclosure statement and competing interests: We updated our journal's competing interests policy in January 2022 and request authors to consider both actual and perceived competing interests. Please review the policy

<https://www.embopress.org/competing-interests> and update your competing interests if necessary.

Please also suggest a striking image or visual abstract to illustrate your article as a PNG file 550 px wide x 300-600 px high. Share synopsis text and image, as well as eTOC:

Please note that these would be the final versions and changes during proofing are usually not allowed

16) As part of the EMBO Publications transparent editorial process initiative (see our policy here:

https://www.embopress.org/transparent-process#Review_Process), Molecular Systems Biology will publish online a Peer Review File (PRF) to accompany accepted manuscripts.

In the event of acceptance, this file will be published in conjunction with your paper and will include the anonymous referee reports, your point-by-point response and all pertinent correspondence relating to the manuscript. Let us know whether you agree with the publication of the PRF and as here, if you want to remove or not any figures from it prior to publication.

Please note that the Authors checklist will be published at the end of the PRF.

Molecular Systems Biology has a "scooping protection" policy, whereby similar findings that are published by others during review or revision are not a criterion for rejection. Should you decide to submit a revised version, I do ask that you get in touch after three months if you have not completed it, to update us on the status.

I look forward to receiving your revised manuscript.

Yours sincerely,

Poonam Bheda, PhD
Scientific Editor
Molecular Systems Biology

Reviewer #1:

In this paper, Jiang and colleagues investigate the possibility that most isoforms produced by alternative splicing or RNA editing could be non-functional noise resulting from imperfections in the splicing and editing machineries.

To do so, they developed a mathematical model to derive expectations about the distribution of isoforms abundances and compare these predictions to the observed distribution for RNA editing in coleoids. They conclude that the observed distribution of RNA editing abundance is consistent with a model where most RNA editing is the result of non-adaptive processes.

One surprising prediction from this model is that, under certain conditions, post-transcriptional modifications can be conserved between lineages, even if these modifications are not advantageous. This is an important point because conservation of editing or alternative splicing is often used as evidence for function. This point deserves to be discussed in more detail.

Overall, I find this study interesting, but somewhat incomplete. A major limitation is that, while the authors develop models for both RNA editing and alternative splicing, they compare their predictions to an actual dataset only in the case of RNA editing. While it might be difficult to collect all the data necessary to perform this comparison with alternative splicing, several papers cited here (references 30 & 32) might provide enough materials for the authors to at least discuss their predictions in the context of alternative splicing.

It is also unclear to me whether a model where most RNA editing is slightly advantageous would really be incompatible with the data in coleoids. While they show that the data on RNA editing in coleoids is consistent with their model when most editing is neutral/slightly deleterious editing noise, I wonder if there would be a distribution of fitness effects where a majority of editing events are very slightly advantageous that would still produce distributions of editing levels compatible with the observed data. I do not recall seeing such a test in the paper.

Finally, even after reading the paper multiple times, I am still a bit unclear as to what the fitness distribution of the editing events is for these models. The authors mention using a Gaussian function for this. However, it seems to me that the most likely distribution of fitness would be one with a few loci where editing is highly advantageous (functional editing), a majority of loci where editing is neutral at low level and possibly trending deleterious at higher levels and a few sites where editing is highly deleterious. A better description of this fitness landscape would help a lot.

Minor point:

Introduction: line 30 replace "essential" with advantageous/useful/important. Essential should be reserved to a function which would result in lethality if it were abolished.

All the data is from one species, which prevents testing the predictions about the impact of effective population size (N_e). It would be nice to see more discussion about which species could be a good place to conduct this test in the future.

Reviewer #2:

The relevance and extent of post-transcriptional regulations in the processes of adaptation is a highly appealing question, asked by many RNA and evolutionary biologists. Authors propose a set of mathematic models describing the processes leading to diverse (m)RNA isoforms, arising mainly by alternative splicing and adenosine-to-inosine editing. They exemplify their models by confirming their previous conclusions on A to I conversion in mRNA in coleoids, an emerging model for the evolution and purpose of RNA editing.

From my point of view (experimental RNA researcher), I can barely assess the arguments supporting the conclusions of the manuscript in its current form. Hence, it is perhaps the weak point of the study. If the data presentation remains as it is, it may not effectively reach the community of experimenters. Accordingly, consider also adequately the actual value of my comments: I felt that the first section defines the system at the organismal level. Therefore, I was puzzled by introducing the alleles in the models (unless just considering di- or polyploidy). (Line 96: What do the authors mean by the number of effector alleles? Similarly, page 6, first line).

Line 107: Reference 56 is a methodological study describing the single-nucleotide resolution of m6A. However, m6A cannot fit for many reasons as a suitable model (such as context-dependence on the mRNA fate or unchanged coding potential).

I associate using the term dysfunctional as the loss-of-function of the product. However, likely most commonly, the modified

(dysfunctional) gene product may not detectably affect the organismal fitness, particularly if produced at low rates (typically, most minor alternative isoforms).

Line 130: Is considering 10^8 generations realistic for most living populations?

Line 213-215: Why should it be the role of just m6A, if there are dozens various modifications in mRNA and lncRNA with largely unknown function? Generally, the second part of this paragraph is way too speculative.

Methods somehow inconsistently accompany the Results and Discussion section.

Reviewer #3:

Jiang et al. use analytical models to investigate evolutionary scenarios that might give rise to alternate gene isoforms and specifically focus on whether deleterious isoforms may arise due to neutral instead of adaptive processes. The question the authors consider is very interesting in my opinion. Much has been done to model the evolution of maladaptive alleles, but I am not aware of this being applied to the evolution of isoform variation. The modeling in this manuscript is well-done, the decisions on the models are fairly well-explained and the results are very much consistent with what I would expect the authors to find. The recapitulation of coleoid RNA editing patterns was a nice finishing touch. Overall, I enjoyed reading this manuscript very much and found the models and discussion led me to consider this process in a novel way. I believe this work would be a good fit for Molecular Systems Biology. I have one major comment where I believe the authors could substantially improve their manuscript and a few minor comments:

Major comment:

I believe the authors could (and probably should) give more discussion of classic models on the interaction of mutation, selection, and drift. Much of this manuscript addresses the potential for neutral evolutionary processes to generate maladaptive isoforms. For example, lines 156-160: "When it is the cis-genotypic value that is examined, results under different values of γ_2 are mostly similar (Fig. S2); when the gene's expression level is high (i.e., optimal P_1 is $\exp(4)$ or $\exp(5)$) and N_e is intermediate (i.e., 103-104), high γ_2 will have a harm-permitting effect: cis-genotypes that lead to production of more I2 will be permitted as the harmful effect is reduced by fast decay of I2 (Fig. S2D-F, red and pink curves)."

If I understand the model correctly, high γ_2 would effectively result in a rapid decay of the maladaptive isoform, thus reducing its overall impact on fitness. This is equivalent to suggesting that an allele (broadly construed) with a very weak negative selection coefficient can exist in a moderate to small population. This result is predicted by both the neutral theory and models of mutation-selection balance, though not specifically about maladaptive isoforms. While this is one example, it does seem like other aspects of this manuscript are also similar to existing models. It would be interesting if the authors could be more clear about where their work is different from previous models of mutation/drift/selection and where this work is an extension of these previous models. I do not believe this is a major problem for this manuscript. Instead, I think this is an opportunity to give more context for the models presented here.

Minor comments:

1: Some of the language used in the manuscript reads a bit oddly to me. For example, in the abstract, the authors questions whether isoforms arise via "mostly adaptive [processes] or the result of noisy molecular processes". I would suggest changing this to adaptive or neutral processes-it is not clear that "noisy molecularly processes" means anything besides neutral evolution and if this is what it means, why not state that? Similarly, the authors frequently suggest that alternate isoforms in their model are non-functional and toxic. Why not just say deleterious as opposed to beneficial?

2: The authors consider editing and splicing mechanisms for generating isoform variation, but do not mention alternate promoters. I believe this is a mistake, as neither of these previous two mechanisms is a good representation for alternate promoters, which are quite common. I do not believe that the authors need to generate new models to account for alternate promoters (as this is not really the goal of the manuscript), but they should probably mention them and give a caveat that further models would need to be generated to account for this isoform class.

Reviewer #1:

In this paper, Jiang and colleagues investigate the possibility that most isoforms produced by alternative splicing or RNA editing could be non-functional noise resulting from imperfections in the splicing and editing machineries.

To do so, they developed a mathematical model to derive expectations about the distribution of isoforms abundances and compare these predictions to the observed distribution for RNA editing in coleoids. They conclude that the observed distribution of RNA editing abundance is consistent with a model where most RNA editing is the result of non-adaptive processes. One surprising prediction from this model is that, under certain conditions, post-transcriptional modifications can be conserved between lineages, even if these modifications are not advantageous. This is an important point because conservation of editing or alternative splicing is often used as evidence for function. This point deserves to be discussed in more detail.

We agree with the reviewers point — we should have made our line of argumentation more explicit. We have updated Results and Discussion to discuss implications of this observation (line 240-244):

“Isoforms that are conserved among species have often been considered as beneficial (e.g., [76]); however, as demonstrated by our results, it is at least plausible that these shared isoforms may be non-adaptive. Therefore, for an individual modification event, phylogenetic conservation does not provide sufficient support for an adaptive hypothesis; to conclude that the modification event under concern is adaptive, further evidence for its biological functions would be necessary.”

Overall, I find this study interesting, but somewhat incomplete. A major limitation is that, while the authors develop models for both RNA editing and alternative splicing, they compare their predictions to an actual dataset only in the case of RNA editing. While it might be difficult to collect all the data necessary to perform this comparison with alternative splicing, several papers cited here (references 30 & 32) might provide enough materials for the authors to at least discuss their predictions in the context of alternative splicing.

This is a fair point. To be frank, we fully acknowledge that the requisite data to test all of the implications of our model is hard to come by. (Indeed, we hope that our paper motivates experimentalists to collect it!). Nonetheless, we agree with the reviewer that it would be useful to engage more with the data that is out there even if we cannot test all the predictions directly. As the reviewer points out, Ref. 32 has presented a correlation between alternative splicing rate and effective population size and we now connect our results to this one. We have updated the text to emphasize the connection between our results and previous empirical studies of alternative splicing (line 188-192):

“It is worth noting that some of these results echo previous findings in empirical studies of various types of gene product diversity. For example, a negative correlation between N_e and the overall rate of alternative splicing has been observed across metazoan species [37]. With N_e

estimates or proxies available from a broad range of metazoans, similar analyses can potentially be done for other types of gene product diversity in the future as more comparative splicing datasets are generated.”

In contrast, there has not been an equivalent analysis for A-to-I editing, which is why we decided to examine A-to-I editing in the last section of Results and Discussion; because data is only available for four species yet the number of editing sites in each species is large, we decided to focus on the distribution of editing levels across editing sites, as explained in line 276-279.

It is also unclear to me whether a model where most RNA editing is slightly advantageous would really be incompatible with the data in coleoids. While they show that the data on RNA editing in coleoids is consistent with their model when most editing is neutral/slightly deleterious editing noise, I wonder if there would be a distribution of fitness effects where a majority of editing events are very slightly advantageous that would still produce distributions of editing levels compatible with the observed data. I do not recall seeing such a test in the paper.

We appreciate the reviewer’s point here. After giving it some careful consideration, we think it is best to exclude this scenario from the revised paper as we do not think it is consistent with what is known about the distribution of fitness effects (DFE) of new mutations. Granted, we know very little about the DFE for mutations that affect editing per se. But there is neither a theoretical or empirical reason to believe that they are substantially different from those of other, better studied mutations. We now discuss this point more explicitly in the revised paper (line 299-315):

“As the effect of A-to-I editing on RNA or protein sequences is equivalent to that of A-to-G mutations, it is intuitive to expect that the distribution of fitness effects (DFE) of A-to-I editing events is similar to that of A-to-G mutations, though the magnitude of fitness effect of editing is likely smaller as each editing event affects only a fraction of transcripts whereas each mutation affects all RNA molecules transcribed from the mutated copy of gene. Although most individual editing events’ fitness effects are unknown, similarity between effects of editing events and mutations that cause the same amino acid change has indeed been shown in empirical studies of editing events with major effects (e.g., [94, 95]). Studies of DFE of non-lethal spontaneous mutations, which are mostly point mutations, have revealed that there are many more deleterious mutations than beneficial ones, and that most of the deleterious mutations have weak effects [96]. Hence, a model where A-to-I editing events are mostly neutral or deleterious is likely to be consistent with the real DFE. Other gene product modifications whose effects on RNA or protein sequences are equivalent to those of point mutations (e.g., C-to-U editing, whose effect resembles that of C-to-T mutations) are also likely to have similar DFEs. The effect of other types of modifications on gene products, on the other hand, are not necessarily comparable to mutations; for example, mis-splicing can result in the inclusion of cryptic (i.e., intronic) sequences in the transcript [6-10] or production of circular RNAs [29]. Nevertheless, as such errors cause even greater disturbance to the gene product’s molecular structure, it is likely they are generally more deleterious than alterations of individual nucleotide

or amino acid sites."

Finally, even after reading the paper multiple times, I am still a bit unclear as to what the fitness distribution of the editing events is for these models. The authors mention using a Gaussian function for this. However, it seems to me that the most likely distribution of fitness would be one with a few loci where editing is highly advantageous (functional editing), a majority of loci where editing is neutral at low level and possibly trending deleterious at higher levels and a few sites where editing is highly deleterious. A better description of this fitness landscape would help a lot.

In this study, Gaussian functions were used to model the relationship between fitness and gene product abundance for individual gene products (a standard, and probably reasonable, assumption in evolutionary models), not for distribution of fitness effects across genes. We apologize for our lack of clarity on this point and we have updated the text accordingly to make this point clear (line 438-442). We have also added discussion about DFE across genes, as mentioned above (line 299-315).

Minor point:

Introduction: line 30 replace "essential" with advantageous/useful/important. Essential should be reserved to a function which would result in lethality if it were abolished.

We agree with the reviewer here and the text has been updated accordingly.

All the data is from one species, which prevents testing the predictions about the impact of effective population size (N_e). It would be nice to see more discussion about which species could be a good place to conduct this test in the future.

We have added that such analyses can be done for metazoans in the future as data of gene product diversity from more species become available (line 190-192). We also mention that such an analysis could be done for A-to-I editing in coleoids if data of editing for more coleoid species and matching N_e estimates become available (line 316-321).

Reviewer #2:

The relevance and extent of post-transcriptional regulations in the processes of adaptation is a highly appealing question, asked by many RNA and evolutionary biologists. Authors propose a set of mathematic models describing the processes leading to diverse (m)RNA isoforms, arising mainly by alternative splicing and adenosine-to-inosine editing. They exemplify their models by confirming their previous conclusions on A to I conversion in mRNA in coleoids, an emerging model for the evolution and purpose of RNA editing.

From my point of view (experimental RNA researcher), I can barely assess the arguments supporting the conclusions of the manuscript in its current form. Hence, it is perhaps the weak

point of the study. If the data presentation remains as it is, it may not effectively reach the community of experimenters. Accordingly, consider also adequately the actual value of my comments:

We appreciate the reviewer's candor here. In writing this paper, our "imagined audience" was both theoreticians and experimentalists (indeed, our author line contains both) and we have tried to strike a balance in terms of technical detail and jargon. We acknowledge that we have not done as good a job as we had hoped and have tried to add some additional context throughout.

I felt that the first section defines the system at the organismal level. Therefore, I was puzzled by introducing the alleles in the models (unless just considering di- or polyploidy). (Line 96: What do the authors mean by the number of effector alleles? Similarly, page 6, first line).

In this manuscript, "alleles" refers to different sequence variants that can potentially exist at a given genomic location (locus) on a chromosome. If a specific sequence variant encodes an RNA or protein sequence motif that is recognized by the modification enzyme such that the RNA or protein molecule could be modified, we would refer to this sequence as an effector allele.

If such modification-facilitating sequence variants are present at multiple loci on the same chromosome, such that the encoded RNA or protein molecule contains modification-facilitating sequence motifs at multiple locations, we will say the genotype (haplotype) contains multiple effector alleles.

We have updated the text of Results and Discussion and of Methods to make the above points more clear (e.g., line 98-111).

Line 107: Reference 56 is a methodological study describing the single-nucleotide resolution of m6A. However, m6A cannot fit for many reasons as a suitable model (such as context-dependence on the mRNA fate or unchanged coding potential).

This is a good point. We have removed this less appropriate example.

I associate using the term dysfunctional as the loss-of-function of the product. However, likely most commonly, the modified (dysfunctional) gene product may not detectably affect the organismal fitness, particularly if produced at low rates (typically, most minor alternative isoforms).

We agree with the reviewer's point, which is also consistent with our finding. Indeed, when the dysfunctional isoform has low abundance, fitness cost it can cause is also low; such a phenotype can indeed be maintained during evolution (Fig. 2, Fig. S1, and Fig. S2).

Line 130: Is considering 10^8 generations realistic for most living populations?

We presented distributions after 10^8 generations to represent a long-term evolutionary outcome. As 10^8 generations is long enough to be considered a macroevolutionary timescale (roughly defined), it is unnecessary to seek an analytical stationary distribution that would be achieved as time approaches infinity. We have added explanations of the reasoning to the text (line 491-493).

Line 213-215: Why should it be the role of just m6A, if there are dozens various modifications in mRNA and lncRNA with largely unknown function? Generally, the second part of this paragraph is way too speculative.

We agree that this paragraph turned out rather speculative, as little is known about most other processes. We have updated the text to make the wording more rigorous.

Methods somehow inconsistently accompany the Results and Discussion section.

We have updated the Results and Discussion to make it neater, keeping only necessary description for methods.

Reviewer #3:

Jiang et al. use analytical models to investigate evolutionary scenarios that might give rise to alternate gene isoforms and specifically focus on whether deleterious isoforms may arise due to neutral instead of adaptive processes. The question the authors consider is very interesting in my opinion. Much has been done to model the evolution of maladaptive alleles, but I am not aware of this being applied to the evolution of isoform variation. The modeling in this manuscript is well-done, the decisions on the models are fairly well-explained and the results are very much consistent with what I would expect the authors to find. The recapitulation of coleoid RNA editing patterns was a nice finishing touch. Overall, I enjoyed reading this manuscript very much and found the models and discussion led me to consider this process in a novel way. I believe this work would be a good fit for Molecular Systems Biology. I have one major comment where I believe the authors could substantially improve their manuscript and a few minor comments:

We thank the reviewer for the encouraging remarks.

Major comment:

I believe the authors could (and probably should) give more discussion of classic models on the interaction of mutation, selection, and drift. Much of this manuscript addresses the potential for neutral evolutionary processes to generate maladaptive isoforms. For example, lines 156-160: "When it is the cis-genotypic value that is examined, results under different values of γ_2 are mostly similar (Fig. S2); when the gene's expression level is high (i.e., optimal P1 is $\exp(4)$ or $\exp(5)$) and N_e is intermediate (i.e., 103-104), high γ_2 will have a harm-permitting effect: cis-

genotypes that lead to production of more I2 will be permitted as the harmful effect is reduced by fast decay of I2 (Fig. S2D-F, red and pink curves)."

If I understand the model correctly, high y_2 would effectively result in a rapid decay of the maladaptive isoform, thus reducing its overall impact on fitness. This is equivalent to suggesting that an allele (broadly construed) with a very weak negative selection coefficient can exist in a moderate to small population. This result is predicted by both the neutral theory and models of mutation-selection balance, though not specifically about maladaptive isoforms. While this is one example, it does seem like other aspects of this manuscript are also similar to existing models. It would be interesting if the authors could be more clear about where their work is different from previous models of mutation/drift/selection and where this work is an extension of these previous models. I do not believe this is a major problem for this manuscript. Instead, I think this is an opportunity to give more context for the models presented here.

We agree with the reviewer's point and have updated the text to put our introduction of the non-adaptive view of gene product diversity and discussions in the context of the nearly neutral theory, including adding the text cited below to Introduction (line 36-42) as well as making other changes:

"Theoretical population genetics have shown that deleterious mutations whose fitness effects are sufficiently mild given the effective population size (N_e) cannot be purged effectively by selection, and can accumulate in the genome over time due to mutations and genetic drift [30-32]. The effect of many molecular errors likely fall into this range, as only a limited fraction of gene product molecules are affected; as a result, selections against mutations that increase error rates can be too weak in small populations to eliminate them in the face of mutational pressure [33, 34]."

Minor comments:

1: Some of the language used in the manuscript reads a bit oddly to me. For example, in the abstract, the authors questions whether isoforms arise via "mostly adaptive [processes] or the result of noisy molecular processes". I would suggest changing this to adaptive or neutral processes-it is not clear that "noisy molecularly processes" means anything besides neutral evolution and if this is what it means, why not state that? Similarly, the authors frequently suggest that alternate isoforms in their model are non-functional and toxic. Why not just say deleterious as opposed to beneficial?

We appreciate the reviewer's point. As noted above, we were trying to strike a balance between an article written for population geneticists/molecular evolution folks and experimentalists. We chose to use terms such as "noisy molecularly processes" and "toxic" to make it clearer that fitness effects of the deleterious isoforms have a mechanistic basis, which we hoped would make it more accessible to the experimentalists. We have updated our text to improve clarity and establish correspondence between terms.

2: The authors consider editing and splicing mechanisms for generating isoform variation, but do not mention alternate promoters. I believe this is a mistake, as neither of these previous two mechanisms is a good representation for alternate promoters, which are quite common. I do not believe that the authors need to generate new models to account for alternate promoters (as this is not really the goal of the manuscript), but they should probably mention them and give a caveat that further models would need to be generated to account for this isoform class.

This is a great point. We agree with the reviewer that the use of alternative promoters cannot be accurately modeled by the models for editing-type or splicing-type modifications. We have added a paragraph to give a caveat (line 140-145):

“It is worth noting that the models of editing-type and splicing-type modifications, while flexible enough for modeling a broad range of processes that generate gene product diversity, may not be well suited for others. For instance, the use of alternative transcription initiation sites can also produce gene product diversity [2-5]. Such diversity cannot be properly modeled as editing-type or splicing type and would require different versions of the model (also see Methods). While the evolutionary dynamics of this type of gene product diversity is beyond the scope of the present paper, it is worth investigation in future studies.”

We have also added a brief description of a modified model that can be applied to the use of alternative promoters (line 381-390).

25th Feb 2025

Manuscript Number: MSB-2024-12620R

Title: Constraints on the optimization of gene product diversity

Dear Dr Pennell,

Thank you for the submission of your revised manuscript to Molecular Systems Biology. I am pleased to inform you that we will be able to accept your manuscript pending the following final amendments:

1) Please rename the "Code and data availability" section to "Data availability" and format the section according to the example below:

"The datasets and computer code produced in this study are available in the following databases:

- Chip-Seq data: Gene Expression Omnibus GSE46748 (<https://www.ncbi.nlm.nih.gov/geo/query/acc.cgi?acc=GSE46748>)
- Modeling computer scripts: GitHub (<https://github.com/SysBioChalmers/GECKO/releases/tag/v1.0>)
- [data type]: [full name of the resource] [accession number/identifier] ([doi or URL or identifiers.org/DATABASE:ACCESSION])"

2) Please include a "Disclosure and competing interests statement". We updated our journal's competing interests policy in January 2022 and request authors to consider both actual and perceived competing interests. Please review the policy <https://www.embopress.org/competing-interests> and update your competing interests if necessary.

3) References: Please correct the reference citation in the reference list to be alphabetical (not numerical) and include the title of the article. Where there are more than 10 authors on a paper, only the first 10 should be listed, followed by "et al.". Please check "Author Guidelines" for more information.

<https://www.embopress.org/page/journal/17574684/authorguide#referencesformat>

4) Please provide separate Results and Discussion sections and remove the Concluding Remarks section.

5) Please place individual sections of the manuscript in the following order: Title page - Abstract & Keywords - Introduction - Results - Discussion - Methods - Data Availability - Acknowledgements - Disclosure and Competing Interests Statement - References - Figure Legends - Expanded View Figure Legends.

6) For the figures and figure legends, please take care of the following:

- Figure titles for figures 1, S5 need to be included in the manuscript.
- The expanded view figures are labelled as supplementary figures in the manuscript. This needs to be rectified using the nomenclature Fig EVX.

- Individual figure legends for figure S3a-c need to be provided in the manuscript.

7) Appendix file: In the Appendix file, the title page should include "Appendix on [manuscript title]" and include a Table of Contents with page numbers of listed items. The nomenclature for appendix figures should be Appendix Figure Sx - please be sure to update the names in the Appendix file and use this nomenclature for callouts in the main manuscript. Every appendix figure should be called out in the main manuscript.

8) Funding: Please ensure that all funding sources are entered into the manuscript submission system. Currently the Jean D'Alembert Foundation (France 2030 program ANR-11-IDEX-0003) is missing.

9) Synopsis:

- Synopsis image: Please provide a graphic that summarises the main findings of the manuscript on a glance and upload it as a high-resolution jpeg file 550 pixels wide x (300-600) pixels high.

- Synopsis text: Please provide a short standfirst (maximum of 300 characters, including space), limit the bullet points to max. 5 and upload it as a separate .doc file. Please write the bullet points to summarise the key NEW findings. They should be designed to be complementary to the abstract - i.e. not repeat the same text. We encourage inclusion of key acronyms and quantitative information (maximum of 30 words / bullet point). Please use the passive voice.

10) As part of the EMBO Publications transparent editorial process initiative (see our policy here:

https://www.embopress.org/transparent-process#Review_Process), Molecular Systems Biology will publish online a Peer Review File (PRF) to accompany accepted manuscripts. This file will be published in conjunction with your paper and will include the anonymous referee reports, your point-by-point response and all pertinent correspondence relating to the manuscript. Let us know whether you agree with the publication of the PRF and as here, if you want to remove or not any figures from it prior to publication. Please note that the Authors checklist will be published at the end of the PRF.

11) After your paper is published, we will promote it on social media. If you have any handles or hashtags for Bluesky you would like included, please let us know.

12) Please provide a point-by-point letter INCLUDING my comments and your detailed responses (as Word file).

I look forward to reading a new revised version of your manuscript as soon as possible.

Yours sincerely,

Poonam Bheda, PhD
Scientific Editor
Molecular Systems Biology

Reviewer #2:

The authors seem to address my specific comments. However, the main concern remains the same. Although I tried my best, the validity of the model leading to proposed conclusions is out of my grasp. Therefore, I decline my review. I apologize to the editors and authors for the inconvenience I may cause.

Reviewer #3:

I thank the authors for taking the time to adjust their manuscript according to my suggestions. I am happy with what they have done and have no further suggestions for them. I believe this manuscript should be published in molecular systems biology.

After reviewing the concerns of reviewer #1 and the response by the authors, I believe that the authors have provided sufficient changes to warrant acceptance. Several of the reviewer's suggestions were included in the manuscript as suggested or by including citations to other previous studies that support their results. The author's response to the concern about the distribution of fitness effects was very reasonable and justified given numerous previous studies of fitness effect distributions for a variety of mutation types. In sum, I believe that the authors have done a satisfactory job in answering this review and, in the absence of concerns directly from reviewer #1, I would recommend acceptance.

We thank the reviewers and the editors for their thoughtful engagement with our work throughout the process. We have edited the text of the paper to split the Results from the Discussion and merged the “Concluding Remarks” section with the Discussion. We have also lightly edited the text throughout for the purposes of readability.

Reviewer #2:

The authors seem to address my specific comments. However, the main concern remains the same. Although I tried my best, the validity of the model leading to proposed conclusions is out of my grasp. Therefore, I decline my review.

I apologize to the editors and authors for the inconvenience I may cause.

We are grateful for your time.

Reviewer #3:

I thank the authors for taking the time to adjust their manuscript according to my suggestions. I am happy with what they have done and have no further suggestions for them. I believe this manuscript should be published in molecular systems biology.

After reviewing the concerns of reviewer #1 and the response by the authors, I believe that the authors have provided sufficient changes to warrant acceptance. Several of the reviewer's suggestions were included in the manuscript as suggested or by including citations to other previous studies that support their results. The author's response to the concern about the distribution of fitness effects was very reasonable and justified given numerous previous studies of fitness effect distributions for a variety of mutation types. In sum, I believe that the authors have done a satisfactory job in answering this review and, in the absence of concerns directly from reviewer #1, I would recommend acceptance.

Thank you for your thoughtful comments.

13th Mar 2025

Manuscript number: MSB-2024-12620RR

Title: Constraints on the optimization of gene product diversity

Dear Dr Pennell,

Thank you again for sending us your revised manuscript. We are now satisfied with the modifications made and I am pleased to inform you that your paper has been accepted for publication.

Yours sincerely,

Sincerely,

Poonam Bheda, PhD
Scientific Editor
Molecular Systems Biology
